# Towards Bridging the gap between Empirical and Certified Robustness against Adversarial Examples

## Abstract

The current state-of-the-art defense methods against adversarial examples typically focus on improving either empirical or certified robustness. Among them, adversarially trained (AT) models produce empirical state-of-the-art defense against adversarial examples without providing any robustness guarantees for large classifiers or higher-dimensional inputs. In contrast, existing randomized smoothing based models achieve state-of-the-art certified robustness while significantly degrading the empirical robustness against adversarial examples. In this paper, we propose a novel method, called *Certification through Adaptation*, that transforms an AT model into a randomized smoothing classifier during inference to provide certified robustness for $\ell_2$ norm without affecting their empirical robustness against adversarial attacks. We also propose *Auto-Noise* technique that efficiently approximates the appropriate noise levels to flexibly certify the test examples using randomized smoothing technique. Our proposed *Certification through Adaptation* with *Auto-Noise* technique achieves an *average certified radius (ACR) scores* up to 1.102 and 1.148 respectively for CIFAR-10 and ImageNet datasets using AT models without affecting their empirical robustness or benign accuracy. Hence, our work is a step towards bridging the gap between the empirical and certified robustness by providing certification using AT models while maintaining the empirical robustness against adversarial examples in the proposed framework.

## 1 Introduction

Deep neural network (DNN) based models are found to be brittle to minor, adversarially-chosen perturbations for their inputs that remain undetectable to human eyes. A DNN classifier that correctly classifies a clean image $x$, can be easily fooled by choosing such *adversarial attacks* to misclassify $x + \delta$ (Szegedy et al., 2014; Goodfellow et al., 2015; Madry et al., 2018). Here, $\delta$ is a minor *adversarial perturbation* such that the change between $x$ and $x + \delta$ remains imperceptible.

Among the existing successful defense models, *adversarial training* (AT) produces the best empirical robustness against the known adversarial attacks, however, without providing any guarantee Madry et al. (2018); Tramèr & Boneh (2019); Zhang et al. (2019); Rice et al. (2020); Gowal et al. (2020). It trains a DNN classifier using strong adversaries from a specific class of perturbation (e.g., a small $\ell_p$-norm) to provide robustness for the same perturbation types. Several certification techniques are proposed that can be applied to adversarially trained models to certifiably verify if the prediction of a test example, $x$ remains constant within its neighborhood Wong & Kolter (2018); Wang et al. (2018); Salman et al. (2019b); Dvijotham et al. (2018); Gehr et al. (2018); Sheikholeslami et al. (2021). However, these certification techniques typically do not scale for larger networks (e.g., ResNet50) and datasets (e.g., ImageNet). Hence, we cannot guarantee for large networks or data-sets that a powerful, not yet known attack would not break these defenses. In fact, several recently proposed empirical defense models are later broken by stronger *adaptive* adversarial attacks, indicating the importance of investigating certified defenses with suitable robustness guarantees Carlini & Wagner (2017); Athalye et al. (2018). In contrast to these models, the *randomized smoothing* based models can provide scalable $\ell_2$-certification framework for any classification model, which is robust against large isotropic Gaussian noise (Cohen et al., 2019; Salman et al., 2019a). However, the existing randomized

| CIFAR-10 models with the best hyper-parameters for $\ell_2$ certifications | | | | | | | | | | | | | | |
|---|---|---|---|---|---|---|---|---|---|---|---|---|---|---|
| Certified $\ell_2$ Radius | 0.0 | 0.25 | 0.5 | 0.75 | 1.0 | 1.25 | 1.5 | 1.75 | 2.0 | 2.25 | 2.5 | 2.75 | 3.0 | ACR |
| Adv$_\infty$ (Rice et al., 2020) | 13.82 | 12.22 | 10.48 | 9.12 | 7.69 | 6.32 | 5.1 | 3.79 | 0.0 | 0.0 | 0.0 | 0.0 | 0.0 | 0.154 |
| Adv$_\infty$ + Auto-Noise | 69.46 | 63.12 | 35.73 | 30.63 | 17.54 | 14.78 | 10.27 | 9.16 | 8.01 | 7.2 | 6.21 | 5.31 | 3.78 | 0.649 |
| Adv$_\infty$ + Adaptation + Auto-Noise | 70.75 | 64.54 | 50.63 | 43.38 | 32.5 | 24.43 | 18.05 | 12.2 | 8.31 | 5.37 | 3.36 | 1.96 | 1.23 | 0.76 |
| Adv$_2$ (Rice et al., 2020) | 30.37 | 26.98 | 23.98 | 21.35 | 18.4 | 15.94 | 13.52 | 10.63 | 0.0 | 0.0 | 0.0 | 0.0 | 0.0 | 0.367 |
| Adv$_2$ + Auto-Noise | 64.45 | 60.57 | 45.73 | 41.06 | 28.48 | 22.92 | 15.1 | 10.77 | 7.23 | 4.8 | 2.77 | 1.67 | 1.1 | 0.702 |
| Adv$_2$ + Adaptation + Auto-Noise | 61.96 | 58.58 | 53.64 | **49.67** | 42.76 | **38.69** | **34.54** | **30.36** | **24.65** | **20.77** | **17.09** | **13.66** | **9.18** | **1.102** |

Table 1: CIFAR-10: Certified accuracy at different $\ell_2$ radii and ACR scores with the best training hyper-parameters (i.e., $\ell_\infty = 12/255$ for Adv$_\infty$ and $\ell_2 = 3.0$ for Adv$_2$). For detailed comparative results with different state-of-the-art methods on both ImageNet and CIFAR-10, please refer to Table 4 and 5 respectively in Appendix. Notably, column $\ell_2 = 0.0$ denotes the performance of the smoothed classifier under Gaussian noise. Our 'certification through adaptation' provides better benign (clean) accuracy by obtaining the predictions directly from the original AT models.

smoothing-based models significantly degrade the empirical robustness compared to the state-of-the-art AT models. In summary, a high empirical robustness along with certification guarantees are necessary to improve the reliability of DNN based frameworks for sensitive real-world applications. However, to the best of our knowledge, none of the existing frameworks provide high performance for both empirical robustness with such certified guarantees using the same DNN classifier.

In this paper, we propose a novel *certification through adaptation* framework that transforms an AT model into a randomized smoothing framework during inference to provide non-trivial $\ell_2$ certification without any additional training or architecture modifications. Our proposed certification technique consists of two steps: We first adapt the AT model using popular batch normalization (BN) adaptation technique with an appropriate levels of Gaussian noise separately for each test example (Cariucci et al., 2017; Li et al., 2016). This process significantly boosts the performance of the AT models against the random isotropic Gaussian noises. Hence, we can now directly apply the *randomized smoothing* based certification technique to provide $\ell_2$ certification in the next step. However, choosing the Gaussian noise for each test example is a challenging task. The existing randomized smoothing based models that use Gaussian noises for training, use the same noise levels to certify each test example, significantly compromising their certification performance. Towards this, we also propose an *Auto-Noise* technique to efficiently approximate the appropriate Gaussian noise levels for correctly certifying each test example during inference. For a given test image, we first obtain the class-prediction using the original AT models (i.e. without adaptation). This step ensures that we can produce the same benign accuracy and empirical robustness for our framework. Next, we certify the predicted class-label using our *certification through adaptation* framework with appropriate noise-level, obtained using *Auto-Noise* technique. We return "ABSTAIN" for test samples where the predicted class from AT classifier is not certifiable. Hence, we provide certification without reducing the same empirical robustness and benign accuracy as the existing AT models (Rice et al., 2020; Madry et al., 2018).

In the following, we summarize the list of contributions for our paper:

**1.** To the best of our knowledge, we are the first to investigate BN adaptation for certification robustness. Our proposed *certification through adaptation* framework is the first to produce non-trivial $\ell_2$ certified robustness from an AT model for large-scale networks (e.g. ResNet) and datasets (e.g. ImageNet). Our proposed technique only requires a set of *clean images*, obtained from training/validation or test set to adapt the AT models for providing the $\ell_2$ certification.

**2.** We also propose *Auto-Noise technique* to efficiently approximate the appropriate Gaussian noise levels for certifying each test example during inference. Auto-Noise is applicable even for existing randomized smoothing based models and often significantly improves the certification performance. Our *Certification through Adaptation* together with *Auto-Noise* technique produces *average certified radius (ACR) scores* upto 1.102 and 1.148 for CIFAR-10 and ImageNet for AT models, achieving the state-of-the-art performance for CIFAR-10. Notably, our proposed method is applied during inference, without affecting the empirical robustness or benign accuracy of AT models to produce these non-trivial $\ell_2$ certification results.

**3.** Our results also suggests a stronger correlation between empirical and certified robustness that empirically stronger AT models also produce better $\ell_2$ certification performance.

## 2 Related Work

### 2.1 Adversarial Robustness for DNN models

#### 2.1.1 Empirical Defenses and Adversarial Training.

Defense models against adversarial examples can be broadly categorized as: *empirical* and *certified* defenses. Empirical defenses demonstrate empirical robustness against adversarial attacks, typically without out providing any certification guarantees (Schott et al., 2019; Moosavi Dezfooli et al., 2019; Nandy et al., 2020; Mao et al., 2021). *Adversarial training* achieves the state-of-the-art empirical defense (Madry et al., 2018). It optimizes the following loss function for a DNN classifier, $f$, to provide robustness within an $\epsilon$-bounded *threat model* for an $\ell_p$ norm, where the perturbations, $\delta \in \Delta$ are constrained as $\Delta = \{\delta : ||\delta||_p \leq \epsilon\}$:

$$\min_\theta \mathbb{E}_{(x,y)}[\max_{\delta \in \Delta} \mathcal{L}(f_\theta(x + \delta), y)] \tag{1}$$

where, $\theta$ denotes the model parameters. $\mathcal{L}$ is the classification loss.

The *inner maximization* in Eq. 1 is solved by producing adversarial examples using strong iterative adversaries, e.g., *projected gradient descent (PGD)* attack (Kurakin et al., 2016; Madry et al., 2018). Wong et al. (2020) found that even a single-step *fast gradient sign method (FGSM)* attack-based AT models also achieves high empirical robustness (Goodfellow et al., 2015). Zhang et al. (2020b) proposed to use the least adversaries for training. Recently TRADES (Zhang et al., 2019), Adv-LLR (Qin et al., 2019) introduced additional regularizers to achieve higher empirical robustness by smoothing the loss surface. Rice et al. (2020) showed that even the standard PGD based AT model with early-stopping criteria provides one of the best empirical defenses for a given perturbation type. Recent works also explored the importance of different hyper-parameters for adversarial training (Gowal et al., 2020; Pang et al., 2021) as well as incorporating additional data in a semi-supervised fashion (Carmon et al., 2019; Uesato et al., 2019) to further improve their empirical robustness against adversarial attacks. Recently, Kireev et al. (2021) also demonstrated that adversarial training with smaller perturbation can also improve the performance against random corruptions.

#### 2.1.2 Certified Defenses.

Empirical defenses demonstrate robustness only against the *known* adversaries without providing any guarantees. In fact, most empirical defenses proposed in the literature were later *broken* by appropriate adversaries, highlighting the importance of certified defenses models (Athalye et al., 2018; Uesato et al., 2018).

Certified defenses provide guarantees that for an input $x$, the classifier's prediction is constant within its neighborhood for a specific class of adversarial perturbation. Several works focus on certifying a trained model by introducing deterministic verification techniques (Tjeng et al., 2017; Gehr et al., 2018; Weng et al., 2018; Wang et al., 2018; Bunel et al., 2018; Zhang et al., 2018; Henriksen & Lomuscio, 2020; Xu et al., 2021; Wang et al., 2021b; Henriksen & Lomuscio, 2021; Palma et al., 2021a;b). Another set of works attempted to train neural network models with provable robustness guarantees, typically using cheaper and incomplete verification techniques. These works include methods based on semidefinite relaxations (Raghunathan et al., 2018), linear relaxations and duality (Wong & Kolter, 2018; Wong et al., 2018), abstract interpretation (Mirman et al., 2018), and interval bound propagation (Gowal et al., 2018). Recently, notable progress are made towards closing the gap between adversarial and provable robustness (Zhang et al., 2020a; Balunovic & Vechev, 2020). Mueller et al. (2021) combined a *small* verification network with a *large*, empirically robust AT model to boost the benign accuracy and empirical robustness of their certified framework. However, these techniques do not scale well for large networks (e.g., ResNet50) or higher-dimensional data (e.g., ImageNet).

#### 2.1.3 Randomized Smoothing.

Randomized smoothing is a promising technique for certifying large-scale networks and higher-dimensional datasets. However, unlike deterministic certification methods, this technique provides robustness certification with probabilistic guarantees. This technique was initially proposed as a heuristic defense (Cao & Gong, 2017; Liu et al., 2018) and later shown to be certifiable (Lecuyer et al., 2019; Li et al., 2019). In the following, we describe the randomized smoothing technique to produce certified $\ell_2$ robustness.

Consider a classification model, $f$, that maps inputs in $\mathbb{R}^d$ to $\mathcal{Y}$ classes. The randomized smoothing framework transforms the original base classifier, $f$ into a new, smoothed classifier $g$ Cohen et al. (2019). In particular, for an input $x \in \mathbb{R}^d$, the smoothed classifier $g$ returns the most probable class to be predicted by the base classifier $f$ under isotropic Gaussian noises of $x$. That is,

$$g(x) = arg \max_{y \in \mathcal{Y}} \mathbb{P}(f(x + \delta) == y) \quad \text{s.t.} \quad \delta \sim \mathcal{N}(0, \sigma^2 I).\tag{2}$$

where, $\sigma^2 I$ is the covariance matrix and $\sigma$ denotes the noise level for certifying $x$. $\sigma$ controls the trade-off between robustness at different $\ell_2$ radii: Increasing $\sigma$ improves the robustness of $g$ at higher $\ell_2$ radii. However, it degrades the robustness at smaller $\ell_2$ radii.

Cohen et al. (2019) presented a tight robustness guarantee using Neyman-Pearson lemma for the smoothed classifier, $g$ and provided an efficient algorithm using Monte Carlo sampling for certification. We can also obtain the same guarantee by explicitly computing the Lipschitz constant of the smoothed classifier as shown in (Salman et al., 2019a; Yang et al., 2020). The certification procedure is as follows: Suppose a base classifier $f$ classifies $\mathcal{N}(x, \sigma^2 I)$ to return the *"most probable"* class, $c_A$ with probability $p_A = \mathbb{P}(f(x + \delta) == c_A)$ and the *"runner-up"* class $c_B$ with probability $p_B = \max_{y \neq c_A} \mathbb{P}(f(x + \delta) == y)$. Then, the smooth classifier, $g$ is certifiably robust around $x$ within an $\ell_2$ radius of $R$, as follows:

$$R = \frac{\sigma}{2} \Big( \Phi^{-1}(p_A) - \Phi^{-1}(p_B) \Big)\tag{3}$$

where, $\Phi^{-1}$ denotes the inverse of the standard Gaussian CDF.

However, computing the exact values for $p_A$ and $p_B$ is impossible in practice when $f$ is a DNN. Cohen et al. (2019) addressed this problem using Monte Carlo sampling to estimate $\underline{p_A}$ and $\overline{p_B}$ such that $\underline{p_A} \leq p_A$ and $\overline{p_B} \geq p_B$ with arbitrarily high probability. The certified radius for input $x$ is then computed by replacing $p_A$ and $p_B$ with $\underline{p_A}$ and $\overline{p_B}$ respectively in Eq. 3.

As we can see in Equation 2 that the base classifier, $f$ needs to be robust against large Gaussian noises to produce non-trivial robustness certification results. Otherwise, it leads to lower $p_A$ and hence a lower certification of $R$ for the test examples. Existing randomized smoothing-based models applies custom-trained using explicit Gaussian noises to learn their original base classifier. Cohen et al. (2019) proposed to train their base-classifier by incorporating random Gaussian noises. Several recent works focused on improving the base classifiers to achieve better certification performance by adversarially choosing the noise (Salman et al., 2019a), incorporating additional regularizers (Zhai et al., 2020; Jeong & Shin, 2020), by ensembling multiple base-models (Horváth et al., 2022) etc. Several works also investigated on improving certification guarantees using different smoothing measures (Li et al., 2019; Lee et al., 2019; Yang et al., 2020) or divergences (Dvijotham et al., 2020). Salman et al. (2020) also demonstrated that we can achieve non-trivial certified robustness even for a standard DNN classifier by incorporating an additional denoising module as a pre-processing unit. Notably, randomized smoothing is the only scalable certification framework and also provides superior performance for different perturbation types (Dvijotham et al., 2020).

However, while achieving the state-of-the-art certification performance, randomized smoothing significantly degrades the empirical robustness against adversarial attacks compared to the state-of-the-art AT models (Lecuyer et al., 2019; Salman et al., 2019a; Cohen et al., 2019). Towards this, our proposed technique transforms an AT model into a randomized smoothing classifier without requiring additional training or architectural modification. Since AT models already provide the state-of-the-art empirical defense, we achieve both empirical and certified robustness against adversarial examples using the same classifier.

### 2.1.4 Batch-normalization and Robustness.

Several recent papers investigate the effects of batch-normalization layers for different aspects of robustness. Many of these works focused on improving robustness against random corruptions by adapting batch-normalization using a sufficiently large set of test images from the same covariate shift (Schneider et al., 2020; Nado et al., 2020; Benz et al., 2021a). By hypothesizing that clean and adversarial examples belongs to different domains, several recent works proposed to apply different branches of BN to separately capture their distributions (Xie et al., 2020a; Xie & Yuille, 2020; Jiang et al., 2020; Wang et al., 2020b; 2021a). Benz

et al. (2021b) presents empirical evidence to argue that BN shifts a model towards being more dependent on non-robust features (NRFs). Unlike these previous works, we proposed to adapt BN layers using appropriate Gaussian noise levels to provide $\ell_2$ certified robustness for AT models.

## 2.2 Test-time Adaptation & applications

Test-time adaptation techniques have been widely explored before in the field of domain adaptation (Sun et al., 2017; Roy et al., 2019; Huang et al., 2018; Li et al., 2016) and covariate-shift adaptation (Sun et al., 2020; French et al., 2017; Xie et al., 2020b; Wang et al., 2020a; Schneider et al., 2020; Nado et al., 2020; Benz et al., 2021a). However, to the best of our knowledge, such techniques are never applied for adversarial robustness and certification. Our paper mainly focuses on one of the most popular and effective mechanisms, called *adaptive batch-normalization.*

A batch-normalization (BN) layer computes the mean and variance of the hidden activation maps across the channels to normalize these activations to $\mathcal{N}(0, 1)$ before feeding into the next hidden layer (Ioffe & Szegedy, 2015). This process reduces the dependencies among different hidden layers, improving the training efficiency for deep architectures. However, the distributional shifts in the test examples lead to different activation statistics compared to the training examples. Hence, the statistics estimated during training fail to correctly normalize the activation tensors to $\mathcal{N}(0, 1)$. Consequently, it breaks the crucial assumption for the subsequent hidden layers to work. Adaptive BN technique computes the BN statistics from the feature activations, $\mu_t$, $s_t^2$, of the test batch. We can adapt them with the existing *training* statistics, $\mu_T$, $s_T^2$, learned using the training batches as (Cariucci et al., 2017; Li et al., 2016; Schneider et al., 2020):

$$\overline{\mu} = \rho \cdot \mu_t + (1 - \rho) \cdot \mu_T \quad \overline{s} = \rho \cdot s_t + (1 - \rho) \cdot s_T \tag{4}$$

where, $\rho \in [0, 1]$ is the momentum. The choice of $\rho = 0$ is equivalent to the standard inference setup with a deterministic DNN classifier in the IID settings. As we receive larger set of samples, we can select *full-adaptation* with $\rho = 1$ to obtain a better estimation of the test distributions.

**Assumptions and Limitations.** The existing BN-adaptation techniques typically require a *large set of test images* from the same "unknown" test distributions. However, this assumption may not hold for several real-world applications, e.g., stateless web APIs. Also, these test images should be *semantically diverse*, preferably over multiple classes, to effectively estimate the test distributions. Hence, it further limits the practical usability of these frameworks for real-world applications, e.g., autonomous cars.

Unlike these models for domain adaptation and corruption robustness, our proposed certification framework does not make any such assumptions. As we shall see that we can appropriately approximate the required Gaussian noise level for adaptation to certify a test image. Therefore, we can pre-select a diverse set of clean images, $\mathbf{X}_{clean}$ and inject the random Gaussian noises to appropriately adapt the models as required, addressing both of these limitations.

## 3 Proposed Methodology

Existing randomized smoothing-based models applies custom-trained using explicit Gaussian noises to learn their original base classifier (Lecuyer et al., 2019; Cohen et al., 2019; Salman et al., 2019a; Zhai et al., 2020; Jeong & Shin, 2020). However, these models produce significantly lower empirical robustness compared to the AT models (Madry et al., 2018; Zhang et al., 2019; Rice et al., 2020; Gowal et al., 2020). In contrast, AT models are not robust against large Gaussian noises in the standard inference settings (Gilmer et al., 2019). In the following, we present *certification through adaptation* with *auto-noise* framework towards bridge this gap between these two research directions by producing certified robustness from AT models.

---

**Algorithm 1:** CERTIFICATION-THROUGH-ADAPTATION $(f, x_{test}, \sigma, \mathbf{X}_{clean}, N)$

---

**Input:** $f$: classifier,   $x_{test}$: test example,   $\sigma$: noise-level,   $\mathbf{X}_{clean}$: batch of clean images sampled from
train/validation data or test stream,   $N$: No. of noisy samples for Monte-Carlo estimation (Eq. 3)
**Output:** Certifiably robust $\ell_2$ radius of $R$ for a single test image, $x_{test}$.

/* (I) Adapt the original classifier $f$ to $f_{adapt}$ with $\rho = 1$ (Eqn 4).                                    */
1  $\tilde{\mathbf{X}} = \{x + \mathcal{N}(0, \sigma I) \ \forall \ x \ \in \ \mathbf{X}_{clean}\}$                               // Perturb $\mathbf{X}_{clean}$ with random noise.
2  $f_{adapt} = \text{BN-ADAPTATION}(f, \tilde{\boldsymbol{X}}, \rho = 1)$                   // Obtain $f_{adapt}$ using BN-adaptation with $\rho = 1$.

/* (II) Certifying $x_{test}$ using randomized smoothing with $f_{adapt}$ as the base classifier.            */
3  $g = \text{GETRANDOMIZEDMODEL}(f_{adapt})$                                    // Convert $f_{adapt}$ to $g$ (Eq 2).
4  $R = \text{CERTIFY}(g, x_{test}; \sigma, N)$                                     // Execute 3 for $\ell_2$ certification.
5  return $\ell_2$ certified radius, $R$

---

### 3.1 Proposed Certification through Adaptation

Given a test image $x_{test}$, our *certification through adaptation* framework consists of two steps: **(I)** adapting the original classifier $f$ to $f_{adapt}$, followed by **(II)** certification using randomized smoothing with $f_{adapt}$ as the base classifier. We summarize our proposed method in Algorithm 1 and describe the steps below:

**(I) Adaptation step:** Recall that adaptive BN requires a large set of diverse test images to correctly re-estimate the BN layer statistics (Cariucci et al., 2017; Li et al., 2016; Schneider et al., 2020). In contrast, for $\ell_2$ certification, we only need to adapt the model, $f$ against Gaussian perturbations using a given noise level $\sigma$ for each $x_{test}$. Hence, unlike existing test-time adaptation-based models for covariate shift or domain adaptation problems, we can pre-select a sufficiently large and diverse set of clean images, $\mathbf{X}_{clean}$ from the training or validation set. In our experiments, we randomly select $1,000$ and $500$ training images for CIFAR-10 and ImageNet respectively.

We first obtain the noisy image-batch, $\tilde{\mathbf{X}} = \{x + \mathcal{N}(0, \sigma I) \ \forall x \in \mathbf{X_{clean}}\}$. Next, with large and diverse noisy image-batch, $\tilde{\mathbf{X}}$, we can apply full BN adaptation using $\rho = 1$ (Eq. 4) and obtain the adapted model as: $f_{adapt} = \text{BN-ADAPTATION}(f, \tilde{\mathbf{X}}, \rho = 1)$.

**(II) certification step:** We use $f_{adapt}$ as the base classifier to obtain the smoothed classifier, $g$. Finally, we execute Eq. 3 to return the $\ell_2$ certified radius for $x_{test}$ using $g$.

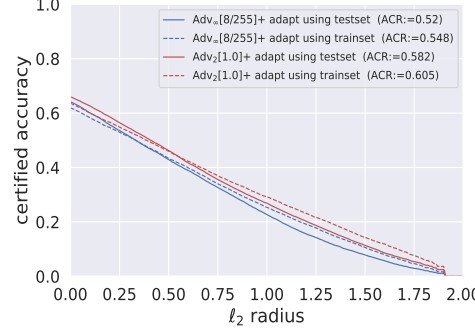

Figure 1: Certification for CIFAR-10 as we apply training data vs test data for adaptation in Algorithm 1. $\text{Adv}_\infty[8/255]$ and $\text{Adv}_2[1.0]$ respectively denote AT models, trained at $\ell_\infty \leq 8/255$ and $\ell_2 \leq 1.0$ threat-boundaries.

**Model adaptation using training versus test images.** We can sample $\mathbf{X}_{clean}$ from training/validation or a stream of test images (if available). Since the underlying distribution of these clean images remains the same, it does not affect the certification performance. Figure 1 verifies this argument. We observe that the certification performance of AT models remains almost the same as we randomly sample $\mathbf{X}_{clean}$ from training vs. test sets. The slight differences in their performances typically arise due to the underlying randomization of $\mathbf{X}_{clean}$ and added random noise to update the BN parameters.

### 3.2 Proposed Auto-Noise: Appropriate Noise level for Certification

Robustness of a classification model can significantly vary at different input spaces. Hence, choosing appropriate noise-level for certifying is an important but challenging task for randomized smoothing based certification techniques. While choosing a lower noise level produces significantly lower-estimates of certified radii, over-estimation of noise may fail to provide any certification robustness for a test example. A brute-force approach to address this problem would be to evaluate the certification results on multiple noise levels and report the maximum certified $\ell_2$ radii. However, certification using randomized smoothing is an extremely time-consuming process: it evaluates a large number of noisy samples (of cardinality $N = 100,000$)

to estimate $\underline{p_A}$ and $\overline{p_B}$ using Monte-Carlo sampling (Eq. 3). For example, it can take upto 110 seconds to certify an ImageNet test example with ResNet-50 models on NVIDIA RTX 2080 Ti (Cohen et al., 2019). Hence, existing randomized smoothing based models typically use the same Gaussian noise level as applied to train their base classifiers (Cohen et al., 2019; Salman et al., 2019a; Zhai et al., 2020; Jeong & Shin, 2020). However, we demonstrate that it significantly underestimates the certification performance of the randomized smoothing framework. Towards this, we present a simple but effective Auto-Noise technique to choose appropriate $\sigma$ for a given test example, $x_{test}$ from a given set of noises, $\boldsymbol{\sigma} = \{\sigma_1, \sigma_2, \cdots\}$:

**Method.** First, we note that randomized smoothing does not require $100,000$ noisy samples to produce $\ell_2$ certified radii. Even a smaller set of noisy examples (e.g. $N_{auto} = 1000$) can provide certification with high probability (e.g. 99.9% confidence), however, for smaller $\ell_2$ certified radii (Cohen et al., 2019). Therefore, it allows us to fairly compare the relative certification of different choices of $\sigma$, separately for each given test image $x_{test}$. We compute the $\ell_2$ certified radii for all $\sigma \in \boldsymbol{\sigma}$ with 99.9% confidence using $N_{auto} = 1000$ noisy samples. Our Auto-Noise technique selects the best $\sigma_{auto}$ that provides the largest $\ell_2$ certification using $N_{auto}$ noisy samples. More formally,

$$\sigma_{auto} = arg \max_{\sigma \in \boldsymbol{\sigma}} \quad \text{CERTIFY}(g, x_{test}; \sigma, N_{auto}) \tag{5}$$

where, $g$ denotes the smoothed classifier obtained using the base classifier, $f$ (Eq. 2). For AT models, we adapt the models, $f_{adapt}$ with noise level $\sigma$ as the base classifier. Finally, we use the best noise level, $\sigma_{auto}$ for certifying $x_{test}$ with a large number of noisy samples, $N = 100,000$. For our certification process, Auto-Noise technique can be incorporated using Algorithm 1 as a sub-routine as shown in Algorithm 2.

---

**Algorithm 2:** CERTIFICATION-THROUGH-ADAPTATION using AUTO-NOISE ($f$, $x_{test}$, $\sigma$, $\mathbf{X}_{clean}$, $N_{auto}$, $N$)

**Input:** $f$: AT classifier, $x_{test}$: test image, $\sigma$: noise list, $\mathbf{X}_{clean}$: clean-image set, $N_{auto}$: No. of noisy samples for Auto-Noise (Eq. 3) $N$: No. of noisy samples for final certification (Eq. 3).

**Output:** Certifiably robust $\ell_2$ radius of $R$ for a single test image, $x_{test}$.

/* Auto-noise step: Execute Algorithm 1 with $N_{auto}$ noisy samples for each $\sigma \in \sigma$. */

1 $\sigma_{auto} = arg \max_{\sigma \in \sigma}$ CERTIFICATION-THROUGH-ADAPTATION($f$, $x_{test}$, $\sigma$, $\mathbf{X}_{clean}$, $N_{auto}$)

/* Algorithm 1 with $N$ number of noisy samples. */

2 $R =$ CERTIFICATION-THROUGH-ADAPTATION($f$, $x_{test}$, $\sigma_{auto}$, $\mathbf{X}_{clean}$, $N$)

3 Return $\ell_2$ certified radius, $R$

---

**Computational Overhead.** In practice, $\sigma_{auto}$ can be chosen from a reasonably small set of noises, $\boldsymbol{\sigma}$ for each test example, $x_{test}$. For example, in our experiments, we select $\boldsymbol{\sigma} = \{0.12, 0.25, 0.37, 0.50, 0.67, 0.75, 0.87, 1.0\}$, i.e. of cardinality=8 and set $N_{auto} = 1,000$. Hence, we require an additional $8,000$ iterations to obtain the appropriate $\sigma$ for each test-examples, along with $100,000$ iterations to get the final certification. In other words, with very little computational overhead, we can approximate the appropriate noise levels for each test example.

Note that our Auto-Noise algorithm using a small set, $N_{auto} = 1,000$ may not provide reliable estimation of the most appropriate noise-level, $\sigma$. However, as shown in Table 1, it significantly improves the certification performance for both AT models and the existing randomized smoothing based models, compared to the fixed choices of $\sigma$ (Cohen et al., 2019; Li et al., 2019). Furthermore, by using certification through adaptation along with Auto-Noise method, we can produce state-of-the-art certification performance for AT models, trained using $\ell_2$ bounded adversarial examples.

**Empirical robustness and benign (clean) accuracy.** Our proposed *certification through adaptation* framework takes an AT model, $f$ and adapt to $f_{adapt}$ to obtain the smooth classifier $g$ to provide $\ell_2$ certification. To maintain the same empirical robustness and benign (clean) accuracy as $f$ for our framework, we obtain class-label prediction from the original classifier, $f$ as the only prediction model. We use $f_{adapt}$ or $g$ only for certification. More formally, for a given test image, $x_{test}$ and ground-truth label $y$:

**Step 1 [Class-Prediction].** Return $f(x_{text})$ as the predicted-class. This step ensures that the benign accuracy and empirical robustness of our framework remains the same as $f$.

**Step 2 [Certification].** Next we obtain $f_{adapt}$ and $g$ from $f$. We only certify $x_{test}$ iff **(a)** the predicted class for both $f$ and $g$ remains the same i.e., $f(x_{test}) == g(x_{test})$ and **(b)** $p_A > 0.5$ (in Eq. 3). Therefore, for a sample $x_{test}$, we may predict the correct class using $f$. However, we may not achieve certified radius using $g$ when $f_{adapt}$ is not robust in the neighborhood of $x_{test}$.

In Table 2 presents the benign accuracy obtained from $f$ and certified accuracy of $g$ at $\ell_2 = 0$. We can see

| Models | Benign Acc. (Using $f$) | Empirical Robustness (Using $f$) | | Certified Acc. at $\ell_2 = 0$ (Using $g$) |
|---|---|---|---|---|
| | | Robust Acc. | Threat boundary | |
| $\text{Adv}_\infty[8/255]$ | 82.6 | 53.4 | $\ell_\infty \leq 8/255$ | 79.69 |
| $\text{Adv}_2[1.0]$ | 83.0 | 54.5 | $\ell_2 \leq 1$ | 82.34 |

Table 2: CIFAR-10: Certification-through-adaptation directly returns the predictions from original AT model, $f$ to maintain the same benign accuracy and empirical robustness. The smoothed classifier, $g$ typically produces lower accuracy at $\ell_2 = 0$.

that $f$ typically produces higher benign accuracy than the certified accuracy of $g$ at $\ell_2 = 0$. However, by incorporating $\sigma = 0$ in the noise-list for Auto-Noise, we can achieve the same benign accuracy as $f$ from $g$.

**Applicability.** Our *certification through adaptation* with *Auto-Noise* method can be applied to any classifier, $f$ with BN layers. However, we cannot improve the robustness of standard non-robust DNN classifiers against large random Gaussian perturbations. Therefore, we can only achieve higher $\ell_2$ certification guarantees at very small $\ell_2$ radii for these models (see in Table 4 and Table 5). Further, we also observe that adapting the existing randomized smoothing models does not necessarily improve the overall certification performance (see Table 1 and Figure 6 in Section 4.3) In contrast, we can significantly improve the performance for AT models against large Gaussian perturbations, providing non-trivial certification robustness.

## 4 Experimental Results

**Setup.** We use CIFAR-10 (Krizhevsky et al., 2009) and ImageNet (Deng et al., 2009) datasets for our experiments. For CIFAR-10, we use pre-activation ResNet18 and for ImageNet, we use ResNet50 (He et al., 2016a;b). For our experiments, we train the AT models using the *early stopping* criteria (Rice et al., 2020). For ImageNet, we use two AT models, $\text{Adv}_\infty[4/255]$ and $\text{Adv}_2[3]$, learned at $\ell_\infty$ and $\ell_2$ threat models with threat boundaries of 4/255 and 3 respectively. For CIFAR-10, we train multiple AT models with different threat boundaries. We denote them by incorporating their corresponding threat boundaries, applied for training. For example, we denote an AT model, trained with threat boundary of 8/255 as $\text{Adv}_\infty[8/255]$.

For our comparisons, we use the standard DNN Baseline and $\text{Rand}_{\sigma=0.5}$ models. Baseline is the standard, non-robust models, trained using clean images. $\text{Rand}_{\sigma=0.5}$ is trained by augmenting random noise, sampled from $\mathcal{N}(0, \sigma^2 I)$ with $\sigma = 0.5$ Cohen et al. (2019). We also compare with the state-of-the-art SmoothAdv models for CIFAR-10 (Salman et al., 2019a). Please refer to Appendix A.3 for additional details [1].

| (a) ImageNet | | | | |
|---|---|---|---|---|
| Model | $\sigma = 0$ | $\sigma = 0.25$ | $\sigma = 0.5$ | $\sigma = 0.75$ |
| Baseline | $\mathbf{75.2}_{\pm0.0}$ | $11.8_{\pm0.22}$ | $0.3_{\pm0.01}$ | $0.1_{\pm0.0}$ |
| + adaptive BN | $74.4_{\pm0.04}$ | $\mathbf{31.0}_{\pm0.27}$ | $\mathbf{7.7}_{\pm.24}$ | $\mathbf{2.4}_{\pm0.01}$ |
| $\text{Adv}_\infty[4/255]$ | $\mathbf{62.8}_{\pm0.0}$ | $3.9_{\pm0.03}$ | $0.4_{\pm0.0}$ | $0.2_{\pm0.01}$ |
| + adaptive BN | $60.8_{\pm0.16}$ | $\mathbf{53.4}_{\pm0.15}$ | $\mathbf{44.9}_{\pm0.08}$ | $\mathbf{33.7}_{\pm0.28}$ |
| $\text{Adv}_2[3]$ | $\mathbf{59.8}_{\pm0.0}$ | $9.8_{\pm0.08}$ | $0.9_{\pm0.01}$ | $0.3_{\pm0.0}$ |
| + adaptive BN | $58.3_{\pm0.08}$ | $\mathbf{53.7}_{\pm0.14}$ | $\mathbf{47.3}_{\pm0.14}$ | $\mathbf{39.8}_{\pm0.18}$ |

| (b) CIFAR-10 | | | | |
|---|---|---|---|---|
| Model | $\sigma = 0$ | $\sigma = 0.25$ | $\sigma = 0.5$ | $\sigma = 0.75$ |
| Baseline | $\mathbf{95.2}_{\pm0.0}$ | $10.9_{\pm0.88}$ | $10.6_{\pm0.76}$ | $10.5_{\pm1.19}$ |
| + adaptive BN | $95.0_{\pm0.57}$ | $\mathbf{40.1}_{\pm0.97}$ | $\mathbf{22.0}_{\pm0.83}$ | $\mathbf{17.2}_{\pm0.66}$ |
| $\text{Adv}_\infty[8/255]$ | $\mathbf{82.1}_{\pm0.0}$ | $40.2_{\pm4.56}$ | $16.1_{\pm7.85}$ | $12.2_{\pm5.23}$ |
| + adaptive BN | $81.6_{\pm0.96}$ | $\mathbf{74.2}_{\pm0.95}$ | $\mathbf{62.4}_{\pm0.64}$ | $\mathbf{51.0}_{\pm1.03}$ |
| $\text{Adv}_2[1]$ | $\mathbf{81.6}_{\pm0.0}$ | $47.5_{\pm5.1}$ | $21.5_{\pm7.79}$ | $14.3_{\pm5.63}$ |
| + adaptive BN | $81.8_{\pm0.7}$ | $\mathbf{75.8}_{\pm0.43}$ | $\mathbf{64.9}_{\pm0.73}$ | $\mathbf{53.5}_{\pm1.71}$ |

Table 3: Top-1 accuracy of AT models significantly improve using adaptive BN under different levels of Gaussian noises (when $\sigma > 0$). We randomly sample the noises and report (*mean $\pm$ 2 $\times$ sd*) for five different runs.

### 4.1 Performance under Gaussian Noise.

We first investigate the performance of AT models as we significantly increase the Gaussian noises. As we note in Equation 2 and 3, it is a necessary condition to provide non-trivial robustness certification at larger $\ell_2$ radii. In Table 3, we present a comparative performance for Baseline, $\text{Adv}_\infty$, and $\text{Adv}_2$ models for both ImageNet and CIFAR-10 datasets. We can see that the classification performance of all these models

---

[1]For ImageNet, we obtain $\text{Adv}_\infty$ and $\text{Adv}_2$ from https://github.com/locuslab/robust_overfitting and Baseline and $\text{Rand}_{\sigma=0.5}$ models from https://github.com/locuslab/smoothing.

sharply degrades under large Gaussian noises in standard inference settings. However, we can improve these performances by adapting them under the same level of Gaussian noises using adaptive BN techniques. In particular, we observe that AT models achieve significantly higher performance gain using adaptive BN than the non-robust, Baseline models under Gaussian noise levels. For example, at $\sigma = 0.5$, Baseline, $\text{Adv}_2[3]$ and $\text{Adv}_\infty[4/255]$ respectively achieve top-1 accuracy of 0.3%, 0.4%, and 0.9% for ImageNet without using BN adaptation (Table 3 (a)). However, adaptive BN for $\text{Adv}_2[3]$ and $\text{Adv}_\infty[4/255]$ significantly improves the top-1 accuracy to 47.3% and 44.9% respectively. In contrast, the baseline model only achieves 7.7% accuracy. We also observe similar results for CIFAR-10 in Table 3 (b).

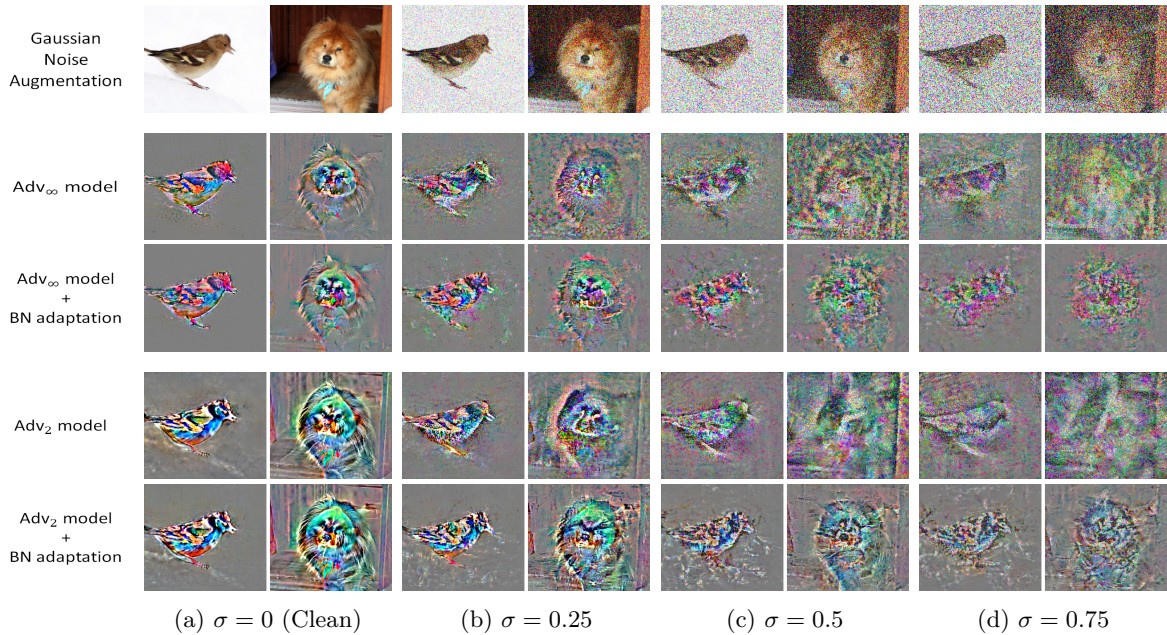

(a) $\sigma = 0$ (Clean)    (b) $\sigma = 0.25$    (c) $\sigma = 0.5$    (d) $\sigma = 0.75$

Figure 2: Visualizing loss-gradients produced by AT models as we apply different levels of Gaussian noises. Additional examples are provided in Figure 10 (Appendix).

**Adaptive BN for AT models correctly extracts robust features under Gaussian noises.** In Figure 2, we further investigate the performance of AT models by visualizing the *loss gradients* of individual pixels of an image as we increase the noise level , $\sigma$. *Loss-gradients* reflect the most relevant input pixels for classification predictions. Here, we scale, translate and clip the loss-gradient values without using any sophisticated techniques (as in Tsipras et al. (2019)). At $\sigma = 0$ (i.e., for clean images), the loss-gradients from AT models align properly with perceptually relevant features (as observed previously (Tsipras et al., 2019; Etmann et al., 2019)). However, as we choose higher noise using $\sigma$=0.5 and $\sigma$=0.75, the overall loss gradients become noisier. Specifically, we can see that AT models without BN adaptation produce larger gradient values (i.e., greater importance) even for background pixels. In contrast, AT models with BN adaptation using Gaussian noises allows to correctly extract perceptually relevant features from the object of interest, suppressing the gradients for background (refer to Figure 2(c) and Figure 2(d)). In other words, it allows us to extract the required semantic information for correct classifications. Also, it is interesting to note that $\text{Adv}_2$ produces significantly more human-aligned loss gradients compared to $\text{Adv}_\infty$. This behavior is also reflected in their classification performance in Table 3 and certification robustness in Table 1. We can see that $\text{Adv}_2$ overall produces better performance compared to $\text{Adv}_\infty$. These results indicate that we can achieve non-trivial certification results by appropriately adapting the AT models.

### 4.2 Certified Robustness for AT models

**Setup.** We now present the comparative $\ell_2$ certification results for AT models using *Certification through adaptation* framewrok. We randomly select $1,000$ and $500$ training images to adapt the AT models for CIFAR-10 and ImageNet respectively (Algorithm 1). We estimate the class-label probabilities of $g$ (Equation

3) using $100,000$ noisy samples and certify the test images with $99.9\%$ confidence, as in (Cohen et al., 2019). We use the entire $10,000$ test images for CIFAR-10 and a sub-sample of $500$ test images for ImageNet.

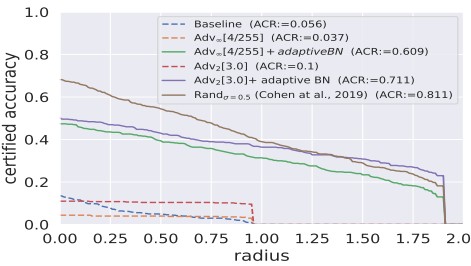 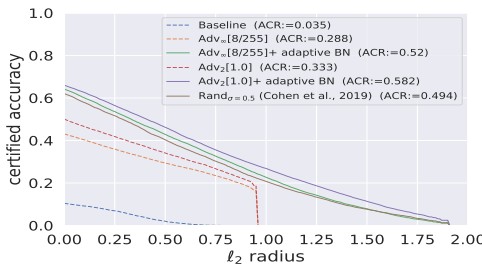

Figure 3: Certification through adaptation produces non-trivial certification at various $\ell_2$ radii for AT models, compared to the non-adapted AT models and Baselines (i.e. standard non-robust DNNs) without adaptation. **(Left)** ImageNet and **(Right)** CIFAR-10.

**Non-trivial certification for AT models.** We note that randomized smoothing technique can be applied to any classifier. However, models that are not robust against large Gaussian noises, produces "trivial" performance i.e., very small $\ell_2$ certified accuracy. In Figure 3, we first demonstrate that AT models can achieve non-trivial $\ell_2$ certified robustness using our proposed certification through adaptation for both ImageNet and CIFAR-10, compare to non-adapted AT models and Baseline without adaptation. Here, we use $\text{Adv}_\infty[4/255]$ and $\text{Adv}_2[3]$ for ImageNet and $\text{Adv}_\infty[8/255]$ and $\text{Adv}_2[1]$ for CIFAR-10. We apply fixed noise levels of $\sigma = 0.5$ to certify all test examples using proposed Algorithm 1. Here, we compare with the certification results of the Baseline, $\text{Adv}_\infty$ and $\text{Adv}_2$ models at $\sigma = 0.25$ in the standard settings (i.e., without adapting these models). We can see a significant boost of $\ell_2$ certification results for both $\text{Adv}_\infty$ and $\text{Adv}_2$ models using our proposed framework. Further, $\text{Adv}_2$ models consistently achieve better certification performance compared to $\text{Adv}_\infty$. We also compare with $\text{Rand}_{\sigma=0.5}$ models at fixed $\sigma = 0.5$, as in Cohen et al. (2019). For CIFAR-10, both $\text{Adv}_\infty[8/255]$ and $\text{Adv}_2[1]$ outperform the $\text{Rand}_{\sigma=0.5}$ models (Cohen et al., 2019). Furthermore, for ImageNet, $\text{Adv}_2[3]$ achieves better certified accuracy compared to $\text{Rand}_{\sigma=0.5}$ beyond $\ell_2$-radii of 1.5. Please refer to Table 4 and 5 (Appendix) for detailed comparisons of different models, trained using different specifications.

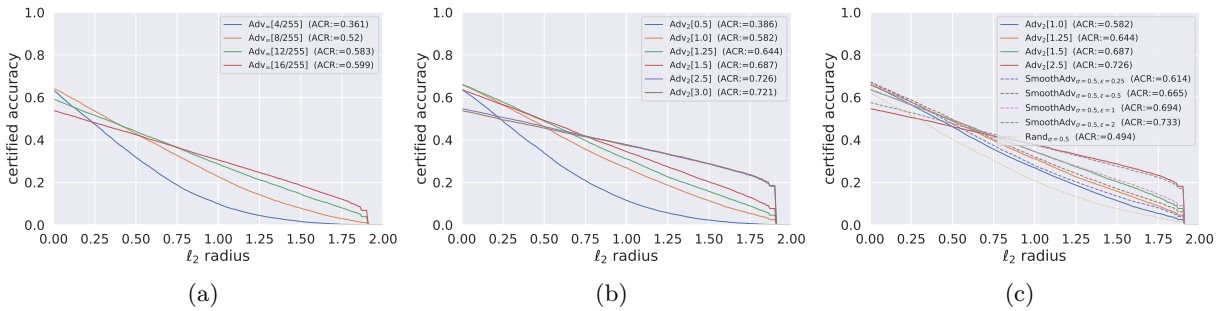

Figure 4: CIFAR-10: (a) $\text{Adv}_\infty$ and (b) $\text{Adv}_2$ models, trained using larger threat boundaries, produces better certification results for higher $\ell_2$ radii and overall larger ACR scores. Here, we apply Algorithm 1 with fixed noise-level $\sigma = 0.5$. (c) Comparing $\text{Adv}_2$ with SmoothAdv (Salman et al., 2019a), trained and certified using $\sigma = 0.5$.

**Larger Threat Boundary for Better Certification.** AT models, trained using a higher threat boundary, produces better certified robustness at higher $\ell_2$ radii as well as larger ACR scores. Figure 4(a) and 4(b) demonstrate this phenomena for CIFAR-10 on both $\text{Adv}_\infty$ and $\text{Adv}_2$ models respectively .

Figure 4(c) also compares the certified accuracy of $\text{Adv}_2$ models with the existing state-of-the-art *SmoothAdv* models (Salman et al., 2019a). SmoothAdv utilizes adversarial training using an adaptive attack with an $\ell_2$ threat boundary of $\epsilon$ and Gaussian noises, $\mathcal{N}(0, \sigma^2 I)$ (See details in Appendix A.3). We set the noise to $\sigma = 0.5$ and vary $\epsilon$ for their training to compare with different SmoothAdv models in Figure 4(c). As we can see that by adapting $\text{Adv}_2$ models at $\sigma = 0.5$ using our proposed Algorithm 1, we can already achieve similar

performance as SmoothAdv in terms of ACR scores. Next, we demonstrate that our *Auto-Noise* technique further improves the performance of both AT models and existing randomized smoothing models.

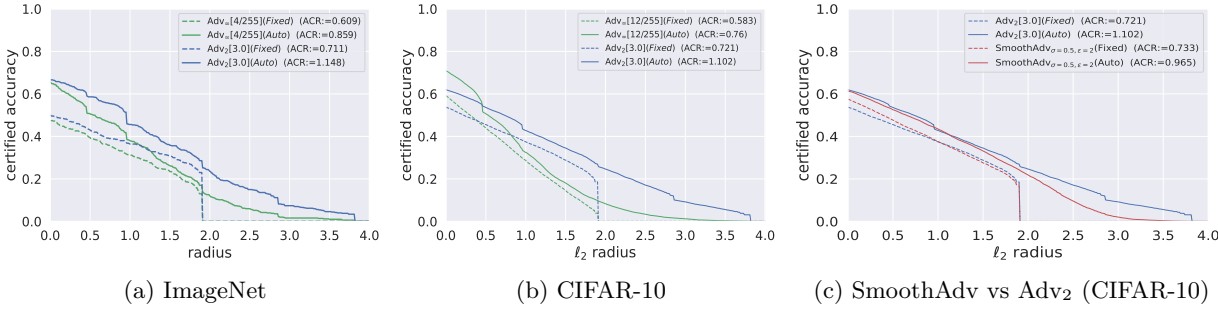

(a) ImageNet        (b) CIFAR-10        (c) SmoothAdv vs $Adv_2$ (CIFAR-10)

Figure 5: Auto-Noise (denoted as "auto") vs. fixed noise at $\sigma = 0.5$ (denoted as "fixed") for $\ell_2$ certification on (a) ImageNet and (b) CIFAR-10 datasets for AT models. (c) SmoothAdv vs $Adv_2$ models for CIFAR-10 using Auto-Noise technique. Here, we only present the results with the best ACR scores.

### 4.3 Auto-Noise: Flexibility of choosing appropriate $\sigma$ for certification.

In Table 3, we can see that the classification models remain robust only for a few test examples under higher Gaussian noise. It suggests that the optimal noise levels for certifying different test examples may vary significantly, indicating the importance of our proposed Auto-Noise technique that efficiently approximate the appropriate noise-level, $\sigma_{auto}$ from $\boldsymbol{\sigma} = \{0.12, 0.25, 0.37, 0.50, 0.67, 0.75, 0.87, 1.0\}$ with $N_{auto} = 1000$.

**Auto-Noise for AT models.**    Figure 5(a) and 5(b) present the certification performance of AT models as we apply both certification through adaptation (Algorithm 1) and Auto-Noise technique for for ImageNet and CIFAR-10 datasets respectively. We compare their performance by certifying using a fixed noise level, $\sigma = 0.5$. We can see that the Auto-Noise technique can significantly improve the performance to achieve 1.148 and 1.102 ACR scores for the best AT models on ImageNet and CIFAR-10 datasets. Figure 5(c) demonstrates that the Auto-Noise technique also improves the performance of the best SmoothAdv model to an ACR score of 0.965 for CIFAR-10. However, our *Certification through Adaptation* with *Auto-Noise* technique for $Adv_2[3]$ model outperforms SmoothAdv on CIFAR-10. In Appendix A.2, we also present the distribution of noise-levels obtained by Auto-Noise technique for different models.

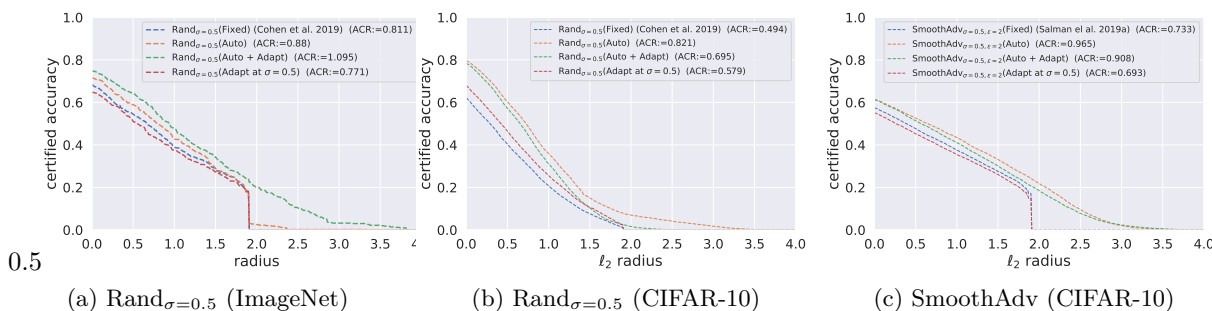

0.5

(a) $Rand_{\sigma=0.5}$ (ImageNet)     (b) $Rand_{\sigma=0.5}$ (CIFAR-10)     (c) SmoothAdv (CIFAR-10)

Figure 6: Effect of adaptation and Auto-Noise technique for existing randomized smoothing based models.

**Certification through Adaptation for existing models.** In Figure 6, we compare the performance of existing randomized smoothing based models as we apply "Certification through Adaptation" with fixed noise-level $\sigma = 0.5$ (i.e. Algorithm 1) and Auto-Noise technique (i.e. Algorithm 2). We can see in Figure 6(a) that our adaptation along with Auto-Noise (Algorithm 2) improves $Rand_{\sigma=0.5}$ model to achieve ACR score of 1.095 on ImageNet dataset. However, our adaptation method alone leads to degrade the overall ACR score from 0.811 to 0.771. In contrast, the certification performance for both $Rand_{\sigma=0.5}$ and SmoothAdv models degrades for CIFAR-10 in presence of BN adaptation using Algorithm 2 (see Figure 6(b) and Figure 6(c)).

Such unusual behavior for these models appear because these models are already trained using Gaussian noise. Hence, further improvement in robustness against different levels of Gaussian noise for a given image becomes challenging by only incorporating BN adaptation techniques.

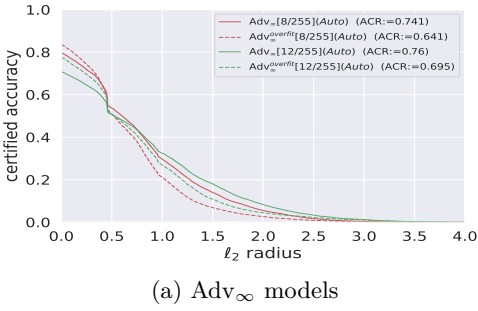

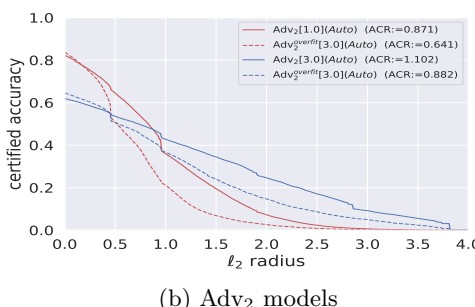

(a) Adv$_\infty$ models

(b) Adv$_2$ models

Figure 7: CIFAR-10: Certification performance degrades for over-fitted AT models when trained without applying early-stopping criteria (Rice et al., 2020). The over-fitted models are denoted as Adv$^{overfit}$.

### 4.4 Over-fitted AT models degrades certification

Rice et al. (2020) demonstrate that AT models *overfit* when trained without *early stopping* criteria. It degrades their empirical robustness against adversarial attacks. Figure 7 compares with the certification results of such *overfitted* AT models, denoted as Adv$^{overfit}$. We observe that Adv$^{overfit}$ models also degrade the certified robustness compared to their corresponding AT models, trained with early stopping criteria. In particular, the difference in their certification performance is more prominent at higher $\ell_2$ radii. Hence, these results (as well as results in Figure 5) indicate that empirical and certified robustness are closely related: *improving the empirical robustness for a model also allows to provide better certified robustness.*

## 5 Conclusion

We propose a novel *certification through adaptation* with Auto-Noise method that automatically selects appropriate noise-levels to adapt the AT models and transform into a smoothed classifier to provide $\ell_2$ certification. Empirically we improve the performance of both AT models and existing randomized smoothing-based models on CIFAR-10 and ImageNet datasets using the Auto-Noise technique. Further, our *Certification through Adaptation* together with *Auto-Noise* technique significantly improves the ACR scores using AT models. Notably, our framework does not affect the empirical robustness or benign accuracy of an AT model to provide these $\ell_2$ certification results.

Several recent methods have significantly improved the state-of-the-art empirical robustness against adversarial attacks (Croce et al., 2021; Mao et al., 2022; Paul & Chen, 2022). Hence, it would be interesting future work to study the interplay between their empirical robustness and certification using the proposed method.

We further note that several modern class of networks have replaced BN with LayerNorm or other instance-based normalization techniques (Ali et al., 2021; Steiner et al., 2022; Liu et al.). Therefore, investigating other adaptation techniques, e.g., self-supervised domain adaptation on single images (Sun et al., 2020), pseudo-labeling (French et al., 2017; Xie et al., 2020b), entropy-minimization (Wang et al., 2020a) etc., is also an useful future study for these modern networks.

### Broader Impact Statement

Improving empirical robustness against adversarial examples along with certification guarantees is an important problem to enhance the reliability of a DNN model for sensitive real-world applications. However, the current state-of-the-art defense methods against adversarial examples typically focus on improving either empirical or certified robustness. In this paper, we aim to bridge this gap by significantly improving the certification performance of AT models without affecting the benign accuracy or reducing their state-of-the-art empirical robustness.

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

# A Appendix

## A.1 Detailed Certification results for different models using proposed method

| ImageNet | | | | | | | | | | | | | | | |
|---|---|---|---|---|---|---|---|---|---|---|---|---|---|---|---|
| Model | Noise-level | $\ell_2$ Radius | | | | | | | | | | | | | |
| | | 0.0 | 0.25 | 0.5 | 0.75 | 1.0 | 1.25 | 1.5 | 1.75 | 2.0 | 2.25 | 2.5 | 2.75 | 3.0 | ACR |
| Baseline | $\sigma = 0.25$ | 13.6 | 7.8 | 4.8 | 3.0 | 0.0 | 0.0 | 0.0 | 0.0 | 0.0 | 0.0 | 0.0 | 0.0 | 0.0 | 0.055 |
| Baseline | Auto-Noise | 52.4 | 36.8 | 5.0 | 3.2 | 0.6 | 0.6 | 0.4 | 0.4 | 0.4 | 0.4 | 0.4 | 0.4 | 0.4 | 0.194 |
| Baseline + Adaptation | Auto-Noise | 57.4 | 25.8 | 4.6 | 1.8 | 0.4 | 0.2 | 0.0 | 0.0 | 0.0 | 0.0 | 0.0 | 0.0 | 0.0 | 0.15 |
| Adv$_\infty$[4/255] | Auto-Noise | 34.6 | 31.2 | 4.0 | 3.8 | 0.6 | 0.6 | 0.4 | 0.4 | 0.2 | 0.2 | 0.0 | 0.0 | 0.0 | 0.169 |
| Adv$_\infty$[4/255] + Adaptation | $\sigma = 0.5$ | 47.4 | 43.6 | 39.4 | 35.8 | 31.4 | 27.6 | 23.4 | 18.2 | 0.0 | 0.0 | 0.0 | 0.0 | 0.0 | 0.609 |
| Adv$_\infty$[4/255] + Adaptation | Auto-Noise | 65.6 | 59.4 | 50.6 | 46.6 | 38.0 | 33.2 | 26.2 | 20.8 | 12.0 | 8.6 | 6.0 | 4.0 | 1.6 | 0.859 |
| Adv$_2$[3.0] | Auto-Noise | 44.0 | 41.6 | 11.8 | 11.2 | 2.8 | 2.6 | 0.4 | 0.4 | 0.0 | 0.0 | 0.0 | 0.0 | 0.0 | 0.261 |
| Adv$_2$[3.0] + Adaptation | $\sigma = 0.5$ | 50.2 | 47.0 | 43.0 | 39.0 | 36.4 | 32.8 | 30.8 | 27.0 | 0.0 | 0.0 | 0.0 | 0.0 | 0.0 | 0.711 |
| Adv$_2$[3.0] + Adaptation | Auto-Noise | 66.6 | 63.8 | 58.6 | 55.4 | 45.6 | **41.0** | **35.8** | **32.4** | **23.6** | **18.6** | **15.0** | **12.8** | **7.4** | **1.148** |
| Rand$_{\sigma=0.5}$ | $\sigma = 0.50$ | 68.2 | 60.8 | 54.4 | 47.8 | 38.8 | 33.8 | 28.6 | 23.4 | 0.0 | 0.0 | 0.0 | 0.0 | 0.0 | 0.811 |
| Rand$_{\sigma=0.5}$ | Auto-Noise | 71.4 | 65.6 | 58.8 | 51.4 | 42.8 | 37.0 | 28.8 | 23.8 | 2.6 | 1.6 | 0.2 | 0.2 | 0.2 | 0.88 |
| Rand$_{\sigma=0.5}$ + Adaptation | Auto-Noise | **74.8** | **69.8** | **64.4** | **56.6** | **47.8** | 40.0 | 34.4 | 27.4 | 20.2 | 15.8 | 10.4 | 6.4 | 3.2 | 1.095 |

Table 4: ImageNet: Certified top-1 accuracy at various $\ell_2$ radii as we vary $\sigma$ for BN adaptation and certification along with average certified radii (ACR). 'Baseline' denotes the standard, non-robust DNN classifier. We use ResNet50 for ImageNet.

| | | ImageNet | | | | | | | | | | | | | |
|---|---|---|---|---|---|---|---|---|---|---|---|---|---|---|---|
| Model | Noise-level | | | | | ℓ₂ Radius | | | | | | | | | |
| | | 0.0 | 0.25 | 0.5 | 0.75 | 1.0 | 1.25 | 1.5 | 1.75 | 2.0 | 2.25 | 2.5 | 2.75 | 3.0 | ACR |
| Baseline | $\sigma = 0.25$ | 10.49 | 6.96 | 2.04 | 0.09 | 0.0 | 0.0 | 0.0 | 0.0 | 0.0 | 0.0 | 0.0 | 0.0 | 0.0 | 0.035 |
| Baseline | Auto-Noise | 33.57 | 18.56 | 10.25 | 4.44 | 0.83 | 0.07 | 0.01 | 0.0 | 0.0 | 0.0 | 0.0 | 0.0 | 0.0 | 0.124 |
| Baseline + Adaptation | Auto-Noise | 59.64 | 21.66 | 7.81 | 3.97 | 1.28 | 0.36 | 0.07 | 0.0 | 0.0 | 0.0 | 0.0 | 0.0 | 0.0 | 0.154 |
| Adv$_\infty$[4/255] | Auto-Noise | 73.71 | 63.04 | 28.75 | 23.91 | 17.96 | 14.19 | 10.12 | 7.08 | 4.99 | 4.14 | 3.24 | 2.49 | 1.84 | 0.57 |
| Adv$_\infty$[4/255] + Adaptation | $\sigma = 0.5$ | 63.23 | 47.34 | 31.83 | 18.78 | 9.98 | 4.44 | 1.62 | 0.28 | 0.0 | 0.0 | 0.0 | 0.0 | 0.0 | 0.361 |
| Adv$_\infty$[4/255] + Adaptation | Auto-Noise | 85.53 | 76.02 | 49.42 | 36.23 | 21.05 | 13.4 | 8.7 | 5.58 | 3.15 | 1.69 | 0.66 | 0.19 | 0.07 | 0.654 |
| Adv$_\infty$[8/255] | Auto-Noise | 74.46 | 66.57 | 36.32 | 30.15 | 18.87 | 13.71 | 8.7 | 5.29 | 2.89 | 1.88 | 1.17 | 0.75 | 0.46 | 0.578 |
| Adv$_\infty$[8/255] + Adaptation | $\sigma = 0.5$ | 64.2 | 53.65 | 42.91 | 32.58 | 22.68 | 14.24 | 7.88 | 2.94 | 0.0 | 0.0 | 0.0 | 0.0 | 0.0 | 0.52 |
| Adv$_\infty$[8/255] + Adaptation | Auto-Noise | 79.69 | 71.78 | 53.76 | 43.61 | 29.68 | 20.87 | 14.04 | 9.08 | 5.53 | 3.33 | 1.71 | 0.81 | 0.35 | 0.741 |
| Adv$_\infty$[12/255] | Auto-Noise | 69.46 | 63.12 | 35.73 | 30.63 | 17.54 | 14.78 | 10.27 | 9.16 | 8.01 | 7.2 | 6.21 | 5.31 | 3.78 | 0.649 |
| Adv$_\infty$[12/255] + Adaptation | $\sigma = 0.5$ | 59.19 | 51.53 | 43.94 | 36.41 | 28.69 | 21.25 | 14.53 | 8.03 | 0.0 | 0.0 | 0.0 | 0.0 | 0.0 | 0.583 |
| Adv$_\infty$[12/255] + Adaptation | Auto-Noise | 70.75 | 64.54 | 50.63 | 43.38 | 32.5 | 24.43 | 18.05 | 12.2 | 8.31 | 5.37 | 3.36 | 1.96 | 1.23 | 0.76 |
| Adv$_\infty$[16/255] | Auto-Noise | 58.47 | 53.41 | 31.48 | 26.76 | 16.68 | 14.04 | 10.89 | 9.17 | 7.42 | 5.62 | 3.81 | 2.43 | 1.53 | 0.55 |
| Adv$_\infty$[16/255] + Adaptation | $\sigma = 0.5$ | 53.8 | 48.07 | 42.51 | 36.54 | 30.55 | 24.68 | 18.49 | 12.11 | 0.0 | 0.0 | 0.0 | 0.0 | 0.0 | 0.599 |
| Adv$_\infty$[16/255] + Adaptation | Auto-Noise | 61.07 | 56.01 | 45.76 | 40.29 | 31.6 | 24.53 | 18.73 | 14.23 | 10.39 | 7.37 | 5.05 | 3.36 | 2.13 | 0.731 |
| Adv$_2$[0.5] | Auto-Noise | 71.6 | 61.2 | 22.17 | 17.64 | 12.26 | 10.97 | 10.17 | 9.76 | 9.48 | 9.12 | 8.53 | 7.52 | 6.61 | 0.603 |
| Adv$_2$[0.5] + Adaptation | $\sigma = 0.5$ | 63.77 | 48.81 | 33.82 | 20.95 | 11.5 | 5.64 | 2.29 | 0.62 | 0.0 | 0.0 | 0.0 | 0.0 | 0.0 | 0.386 |
| Adv$_2$[0.5] + Adaptation | Auto-Noise | 86.26 | 77.52 | 61.25 | 46.44 | 23.42 | 16.22 | 11.55 | 9.2 | 7.62 | 6.47 | 5.07 | 3.81 | 2.51 | 0.796 |
| Adv$_2$[1.0] | Auto-Noise | 81.08 | 71.0 | 43.82 | 34.77 | 17.25 | 11.15 | 6.19 | 3.68 | 1.76 | 1.02 | 0.52 | 0.27 | 0.15 | 0.608 |
| Adv$_2$[1.0] + Adaptation | $\sigma = 0.5$ | 66.05 | 56.45 | 46.24 | 35.6 | 26.89 | 18.73 | 11.37 | 5.41 | 0.0 | 0.0 | 0.0 | 0.0 | 0.0 | 0.582 |
| Adv$_2$[1.0] + Adaptation | Auto-Noise | 82.34 | 75.38 | 64.53 | 53.8 | 36.04 | 27.55 | 19.53 | 12.61 | 7.29 | 4.33 | 2.39 | 1.21 | 0.67 | 0.871 |
| Adv$_2$[1.25] | Auto-Noise | 78.32 | 72.56 | 44.86 | 39.13 | 25.98 | 21.02 | 16.31 | 12.17 | 8.18 | 5.18 | 2.25 | 1.25 | 0.66 | 0.741 |
| Adv$_2$[1.25] + Adaptation | $\sigma = 0.5$ | 66.22 | 57.73 | 48.8 | 39.64 | 31.07 | 22.61 | 15.82 | 8.96 | 0.0 | 0.0 | 0.0 | 0.0 | 0.0 | 0.644 |
| Adv$_2$[1.25] + Adaptation | Auto-Noise | 80.52 | 74.25 | 64.95 | 56.56 | 40.81 | 32.71 | 24.96 | 17.87 | 11.5 | 8.07 | 5.88 | 4.05 | 2.59 | 0.972 |
| Adv$_2$[1.5] | Auto-Noise | 75.26 | 69.75 | 48.42 | 41.73 | 26.44 | 20.82 | 14.12 | 10.14 | 6.87 | 4.75 | 3.02 | 1.81 | 0.87 | 0.733 |
| Adv$_2$[1.5] + Adaptation | $\sigma = 0.5$ | 63.67 | 56.55 | 49.19 | 41.72 | 34.47 | 27.36 | 20.23 | 12.98 | 0.0 | 0.0 | 0.0 | 0.0 | 0.0 | 0.687 |
| Adv$_2$[1.5] + Adaptation | Auto-Noise | 76.22 | 70.47 | 62.39 | 55.91 | 42.45 | 35.79 | 29.01 | 21.71 | 14.23 | 10.04 | 6.51 | 4.07 | 2.29 | 0.99 |
| Adv$_2$[2.5] | Auto-Noise | 61.2 | 57.42 | 42.26 | 38.0 | 26.56 | 21.41 | 14.21 | 10.34 | 7.33 | 4.96 | 3.24 | 2.06 | 1.07 | 0.662 |
| Adv$_2$[2.5] + Adaptation | $\sigma = 0.5$ | 54.73 | 50.53 | 46.26 | 41.84 | 37.74 | 33.2 | 28.69 | 23.34 | 0.0 | 0.0 | 0.0 | 0.0 | 0.0 | 0.726 |
| Adv$_2$[2.5] + Adaptation | Auto-Noise | 63.36 | 59.53 | 54.68 | 50.3 | 42.95 | 38.62 | 33.97 | 29.42 | 23.03 | 19.22 | 15.3 | 11.34 | 7.51 | 1.073 |
| Adv$_2$[3.0] | Auto-Noise | 64.45 | 60.57 | 45.73 | 41.06 | 28.48 | 22.92 | 15.1 | 10.77 | 7.23 | 4.8 | 2.77 | 1.67 | 1.1 | 0.702 |
| Adv$_2$[3.0] + Adaptation | $\sigma = 0.5$ | 53.75 | 49.41 | 45.57 | 41.52 | 37.43 | 33.37 | 28.82 | 23.65 | 0.0 | 0.0 | 0.0 | 0.0 | 0.0 | 0.721 |
| Adv$_2$[3.0] + Adaptation | Auto-Noise | 61.96 | 58.58 | 53.64 | 49.67 | 42.76 | 38.69 | 34.54 | 30.36 | 24.65 | 20.77 | 17.09 | 13.66 | 9.18 | 1.102 |
| Rand$_{\sigma=0.5}$ | $\sigma = 0.5$ | 62.13 | 51.68 | 40.38 | 30.25 | 20.81 | 13.36 | 7.71 | 3.38 | 0.0 | 0.0 | 0.0 | 0.0 | 0.0 | 0.494 |
| Rand$_{\sigma=0.5}$ | Auto-Noise | 79.48 | 71.75 | 60.23 | 48.72 | 35.97 | 25.16 | 15.09 | 10.13 | 6.98 | 5.58 | 4.18 | 2.94 | 1.76 | 0.821 |
| Rand$_{\sigma=0.5}$ + Adaptation | Auto-Noise | 78.27 | 69.51 | 57.16 | 44.73 | 31.19 | 19.27 | 10.4 | 4.25 | 1.65 | 0.57 | 0.14 | 0.01 | 0.0 | 0.695 |
| SmoothAdv$_{\sigma=0.5,\epsilon=0.25}$ | $\sigma = 0.5$ | 67.35 | 57.8 | 47.63 | 37.41 | 27.88 | 20.33 | 13.53 | 8.03 | 0.0 | 0.0 | 0.0 | 0.0 | 0.0 | 0.614 |
| SmoothAdv$_{\sigma=0.5,\epsilon=0.25}$ | Auto-Noise | 72.88 | 65.23 | 55.25 | 44.87 | 34.93 | 24.79 | 16.44 | 9.17 | 4.92 | 1.91 | 0.73 | 0.21 | 0.07 | 0.737 |
| SmoothAdv$_{\sigma=0.5,\epsilon=0.25}$ + Adaptation | Auto-Noise | 74.45 | 65.68 | 54.28 | 42.59 | 31.76 | 21.61 | 13.19 | 7.16 | 3.41 | 1.43 | 0.52 | 0.16 | 0.05 | 0.697 |
| SmoothAdv$_{\sigma=0.5,\epsilon=0.5}$ | $\sigma = 0.5$ | 67.21 | 58.82 | 49.68 | 40.35 | 31.93 | 24.18 | 17.05 | 10.57 | 0.0 | 0.0 | 0.0 | 0.0 | 0.0 | 0.665 |
| SmoothAdv$_{\sigma=0.5,\epsilon=0.5}$ | Auto-Noise | 71.85 | 65.28 | 56.41 | 48.2 | 39.47 | 29.82 | 21.44 | 13.55 | 7.54 | 3.45 | 1.35 | 0.41 | 0.12 | 0.807 |
| SmoothAdv$_{\sigma=0.5,\epsilon=0.5}$ + Adaptation | Auto-Noise | 73.54 | 66.14 | 56.81 | 47.04 | 37.2 | 26.99 | 18.46 | 10.78 | 5.67 | 2.6 | 1.0 | 0.44 | 0.13 | 0.773 |
| SmoothAdv$_{\sigma=0.5,\epsilon=1.0}$ | $\sigma = 0.5$ | 63.95 | 56.53 | 49.53 | 41.38 | 34.63 | 27.81 | 21.22 | 14.41 | 0.0 | 0.0 | 0.0 | 0.0 | 0.0 | 0.694 |
| SmoothAdv$_{\sigma=0.5,\epsilon=1.0}$ | Auto-Noise | 67.91 | 62.28 | 55.36 | 48.41 | 41.14 | 33.57 | 26.27 | 18.42 | 11.87 | 6.1 | 2.64 | 0.95 | 0.32 | 0.854 |
| SmoothAdv$_{\sigma=0.5,\epsilon=1.0}$ + Adaptation | Auto-Noise | 69.6 | 63.04 | 55.52 | 47.65 | 38.93 | 30.7 | 22.57 | 15.34 | 9.21 | 4.51 | 2.02 | 0.9 | 0.33 | 0.813 |
| SmoothAdv$_{\sigma=0.5,\epsilon=2.0}$ | $\sigma = 0.5$ | 57.59 | 52.82 | 47.67 | 42.68 | 37.55 | 32.64 | 27.52 | 22.42 | 0.0 | 0.0 | 0.0 | 0.0 | 0.0 | 0.733 |
| SmoothAdv$_{\sigma=0.5,\epsilon=2.0}$ | Auto-Noise | 61.27 | 57.27 | 52.52 | 48.17 | 43.49 | 38.02 | 33.15 | 27.47 | 21.86 | 15.81 | 9.5 | 4.97 | 2.01 | 0.965 |
| SmoothAdv$_{\sigma=0.5,\epsilon=2.0}$ + Adaptation | Auto-Noise | 61.23 | 56.9 | 51.33 | 46.44 | 41.05 | 35.65 | 30.11 | 24.35 | 18.48 | 12.8 | 7.76 | 4.27 | 2.3 | 0.908 |

Table 5: CIFAR-10: Certified top-1 accuracy at various $\ell_2$ radii as we vary $\sigma$ for test-time BN adaptation along with average certified radii (ACR) for individual settings. 'Baseline' denotes the standard, non-robust DNN classifier. We use pre-activation ResNet-18 model for CIFAR-10.

## A.2 Distribution of $\sigma_{auto}$ generated by Auto-Noise Method.

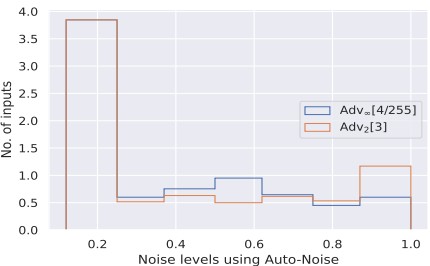 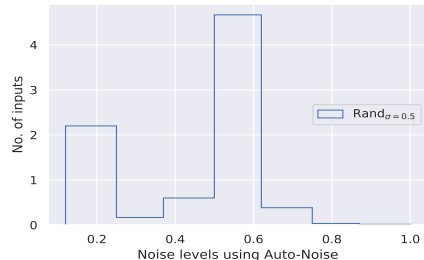

Figure 8: ImageNet: Visualizing the distribution of noise-level produced by Auto-Noise technique for the 500 sub-sampled images. (Left) AT models, (Right) Rand$_{\sigma=0.5}$ without adaptation.

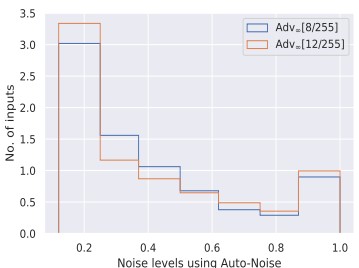 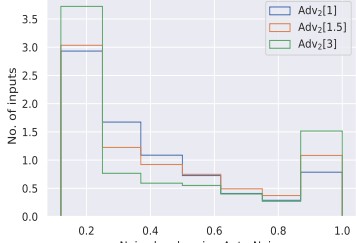 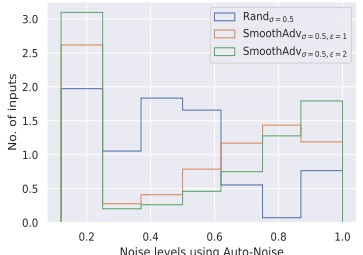

Figure 9: CIFAR-10: Visualizing the distribution of noise-level produced by Auto-Noise technique for the entire $10,000$ test images. (Left) Adv$_\infty$ models, (Middle) Adv$_2$ models (Right) existing randomized smoothing models without adaptation.

### A.3 Implementation Details

We present our experimental results on CIFAR-10 (Krizhevsky et al., 2009) and ImageNet (Deng et al., 2009) datasets. The descriptions of different models and training hyper-parameters are provided in the following:

#### A.3.1 CIFAR-10.

We use pre-activation ResNet18 architecture (He et al., 2016b) for our experiments on CIFAR-10. We apply the SGD optimizer with a batch size of 128. We execute a total of 200 training epochs and apply a step-wise learning rate decay set initially at 0.1 and divided by 10 at 100 and 150 epochs, and weight decay $5 \times 10^{-4}$.

**AT models (Madry et al., 2018; Rice et al., 2020):** Unless and otherwise specified, our AT models are learned using early stopping criteria as described in (Rice et al., 2020). We learn several AT models with different threat boundaries for our experiments. We denote them by specifying their corresponding threat model and threat boundaries. For example, Adv$_2$[1.5] denotes an AT model that is learned using PGD adversary with $\ell_2$ threat model and a threat boundary of $\epsilon = 1.5$, along with *early-stopping criteria* (Rice et al., 2020). We also learn AT models *without* using early-stopping criteria, as in (Madry et al., 2018) for our comparison in Figure 7. These models are denoted as Adv$^{overfit}$.

We use *projected gradient descent (PGD)* adversarial attack (Madry et al., 2018) to train these AT models as follows: For Adv$_\infty$, we use 10 iterations and an $\ell_\infty$ step size of $\epsilon/4$. For Adv$_2$, we use 10 iterations and an $\ell_2$ step size of $\epsilon/8.5$. This is the same experimental setup as in (Rice et al., 2020)). We choose a small set of $1,000$ images from the CIFAR-10 test set for our validation. We apply the PGD attack with the same hyper-parameters for our validation during training. We save the best model using the *early-stopping* criteria (Rice et al., 2020).

**Randomized smoothing model by Cohen et al. (2019):** We also train Rand$_{\sigma=0.5}$ by training with augmented random noise, sampled from an isotropic Gaussian distribution $\mathcal{N}(0, \sigma^2 I)$ with $\sigma = 0.5$. Here, we

keep the same model architecture, learning rates, batch sizes, and other hyper-parameters as used to learn the AT models.

**Randomized smoothing model by Salman et al. (2019a):** We also compare with the state-of-the-art certification models, called 'SmoothAdv', by Salman et al. (2019a) for our experiments on $\ell_2$ certification We train the SmoothAdv models by choosing random noise vectors followed by an adaptive adversarial attack with specified $\ell_2$ threat boundary of $\epsilon$ at each iteration. The noise vectors are sampled from an isotropic Gaussian distribution $\mathcal{N}(0, \sigma^2 I)$.

We note that the training hyper-parameter $\epsilon$ has the most significant impact on the certification curve for a SmoothAdv model (please refer to Table 7-15 of (Salman et al., 2019a) for more details). For our experiments, we train 4 different SmoothAdv models with $\epsilon = \{0.25, 0.5, 1, 2\}$ and $\sigma = 0.5$ using adaptive PGD attack with 10 steps. We denote them as SmoothAdv$_{\sigma=0.5, \epsilon=0.25}$, SmoothAdv$_{\sigma=0.5, \epsilon=0.5}$, SmoothAdv$_{\sigma=0.5, \epsilon=1}$ and SmoothAdv$_{\sigma=0.5, \epsilon=2}$ respectively. We use the same training set-up and other hyper-parameters as specified in their Github: https://github.com/Hadisalman/smoothing-adversarial.

### A.3.2 ImageNet.

We use ResNet50 architecture (He et al., 2016a) for ImageNet. We obtain the Baseline and Rand$_{\sigma=0.5}$ models from (Cohen et al., 2019)[2]. These models are trained using Gaussian augmented noises, sampled from isotropic Gaussian distribution $\mathcal{N}(0, \sigma^2 I)$ with $\sigma = 0.0$ (i.e., no noise) and $\sigma = 0.5$ respectively.

The AT models i.e., Adv$_\infty$[4/255] and Adv$_2$[3.0] are learned for $\ell_\infty$ and $\ell_2$ threat models with threat boundary of 4/255 and 3, respectively. We use the publicly available models provided by Rice et al. (2020) [3]. These models are fine-tuned using PGD-based adversarial training with early stopping criteria, originally provided by Engstrom et al. (2019) [4].

We resize the input images to $256 \times 265$ pixels and crop $224 \times 224$ pixels from the center. For our experiments on certification, we use a set of 500 test images by choosing at most 1 sample for each class.

### A.4 Choice of Adaptive BN hyper-parameters

BN adaptation technique is controlled by two hyper-parameters, i.e., the *test batch-size* and *momentum* ($\rho$) (see Equation 4) to update the statistics of the batch-normalization layers. Assuming that the test images are obtained independently from the same test distribution, we can efficiently compute the BN statistics from these images. The hyper-parameter $\rho \in [0, 1]$ controls the tread-off between pre-computed training statistics and test statistics. We can obtain a better estimation of the test distribution from a large test batch. Hence, we can choose a higher value of $\rho$.

Here, we compare the top-1 test accuracy of AT models under Gaussian augmented noise with $\sigma = 0.5$ for different choices of $\rho$ and the batch size. We skip the standard baseline models from our analysis and refer to the previous works (Schneider et al., 2020; Nado et al., 2020) that analyzed the effects of these hyper-parameters for the standard baseline DNN classifiers.

**Momentum ($\rho$).** We first investigate the effect of momentum ($\rho$) as we choose a large batch size of 512. In Table 6, we present the performance of AT models for different values of $\rho$. Recall that, $\rho = 1$ denotes *full adaptation* (Equation 4). Here, we completely ignore the training statistics and recompute the BN statistics using the test batches. In contrast, $\rho = 0$ represents *no adaptation*, i.e., the same as the standard 'deterministic' inference setup. In this case, we use the previously computed BN statistics obtained during training.

We observe that for ImageNet (Table 6 [Left]) the performance started converging at $\rho = 0.7$. For CIFAR-10 (Table 6 [Right]), the convergence started even earlier at $\rho = 0.5$.

---

[2]https://github.com/locuslab/smoothing
[3]https://github.com/locuslab/robust_overfitting
[4]https://github.com/MadryLab/robustness

| (a) ImageNet | | | | (b) CIFAR-10 | | |
|---|---|---|---|---|---|---|
| $\rho$ | Adv$_\infty$ | Adv$_2$ | | $\rho$ | Adv$_\infty$ | Adv$_2$ |
| 0.0 (No adaptation) | $0.4_{\pm 0.01}$ | $0.9_{\pm 0.01}$ | | 0.0 (No adaptation) | $16.1_{\pm 7.85}$ | $21.5_{\pm 7.79}$ |
| 0.1 | $2.1_{\pm 0.04}$ | $7.7_{\pm 0.09}$ | | 0.1 | $45.1_{\pm 0.49}$ | $46.9_{\pm 0.48}$ |
| 0.3 | $20.6_{\pm 0.16}$ | $36.6_{\pm 0.09}$ | | 0.3 | $59.2_{\pm 0.42}$ | $60.8_{\pm 0.33}$ |
| 0.5 | $41.1_{\pm 0.09}$ | $45.5_{\pm 0.13}$ | | 0.5 | $62.4_{\pm 0.27}$ | $64.4_{\pm 0.6}$ |
| 0.7 | $43.5_{\pm 0.14}$ | $46.7_{\pm 0.13}$ | | 0.7 | $62.8_{\pm 0.52}$ | $64.9_{\pm 0.31}$ |
| 0.9 | $44.2_{\pm 0.12}$ | $46.8_{\pm 0.13}$ | | 0.9 | $62.8_{\pm 0.71}$ | $64.9_{\pm 0.31}$ |
| 1.0 (Full adaptation) | $44.8_{\pm 0.13}$ | $47.2_{\pm 0.14}$ | | 1.0 (Full adaptation) | $62.4_{\pm 0.64}$ | $64.9_{\pm 0.73}$ |

Table 6: Top-1 accuracy using fixed test batch-size $= 512$ for AT models under Gaussian augmented noise with $\sigma = 0.5$ for different choices of momentum, $\rho$ during inference. We randomly shuffle the test images to report $(mean + 2 \times sd)$ of 5 different runs.

| (a) ImageNet | | | | (b) CIFAR-10 | | |
|---|---|---|---|---|---|---|
| Batch Size | Adv$_\infty$ | Adv$_2$ | | Batch Size | Adv$_\infty$ | Adv$_2$ |
| w/o BN adapt | $0.4_{\pm 0.01}$ | $0.9_{\pm 0.01}$ | | w/o BN adapt | $16.1_{\pm 7.85}$ | $21.5_{\pm 7.79}$ |
| 8 | $11.5_{\pm 0.22}$ | $9.1_{\pm 0.15}$ | | 8 | $57.2_{\pm 1.23}$ | $59.5_{\pm 0.38}$ |
| 16 | $28.1_{\pm 0.22}$ | $26.7_{\pm 0.14}$ | | 16 | $60.2_{\pm 0.79}$ | $62.3_{\pm 0.87}$ |
| 32 | $37.1_{\pm 0.24}$ | $37.6_{\pm 0.2}$ | | 32 | $61.5_{\pm 0.46}$ | $63.6_{\pm 0.55}$ |
| 64 | $41.4_{\pm 0.26}$ | $42.9_{\pm 0.12}$ | | 64 | $62.3_{\pm 0.5}$ | $64.0_{\pm 0.38}$ |
| 128 | $43.3_{\pm 0.15}$ | $45.4_{\pm 0.13}$ | | 128 | $62.7_{\pm 0.68}$ | $64.4_{\pm 0.53}$ |
| 256 | $44.4_{\pm 0.21}$ | $46.7_{\pm 0.07}$ | | 256 | $62.7_{\pm 0.68}$ | $64.9_{\pm 0.48}$ |
| 512 | $44.8_{\pm 0.13}$ | $47.2_{\pm 0.14}$ | | 512 | $62.4_{\pm 0.64}$ | $64.9_{\pm 0.73}$ |

Table 7: Top-1 accuracy using fixed $\rho = 1$ for AT models under Gaussian augmented noise with $\sigma = 0.5$ for different size of test batches during inference. We randomly shuffle the test images to report $(mean + 2 \times s.d.)$ of 5 different runs.

**Batch Size.**    Next, we investigate the minimum size of the test batches to choose $\rho = 1$ (i.e., full-adaptation). In Table 7, we fix $\rho = 1$ and vary the test batch sizes as we evaluate these AT models. We observe that the performance of these models started improving even when we are using the test batches of size 8. The performance further improves as we choose larger sizes of test batches. We can see that their performance started converging as we choose the test batches of size 64 for ImageNet. On the other hand, the convergence started much earlier for CIFAR-10.

## A.5    Performance against different corruptions

We mainly focus on $\ell_2$ certification using Gaussian noise in this paper. However, we note that randomized smoothing techniques have been also applied to provide certifications for other perturbation types as well (e.g., random uniform noise for $\ell_1$ norm (Yang et al., 2020)). Consequently, we can apply our proposed Algorithm 1 to adapt an AT model for any given perturbation types without any additional training for different applications.

Further, Hendrycks & Dietterich (2019) recently introduced ImageNet-C and CIFAR10-C datasets by *algorithmically generated random corruptions* from *noise*, *blur*, *weather*, and *digital* categories with 5 different severity levels for each corruption. Several recent works demonstrated that adaptive BN techniques can significantly improve the performance of any classifier (including AT models) against different random corruptions. Further, – also demonstrated the effectiveness of AT models even without applying any adaptation. Hence, our proposed certification framework for AT models is a step forward towards further improving the reliability of sensitive real-world applications.

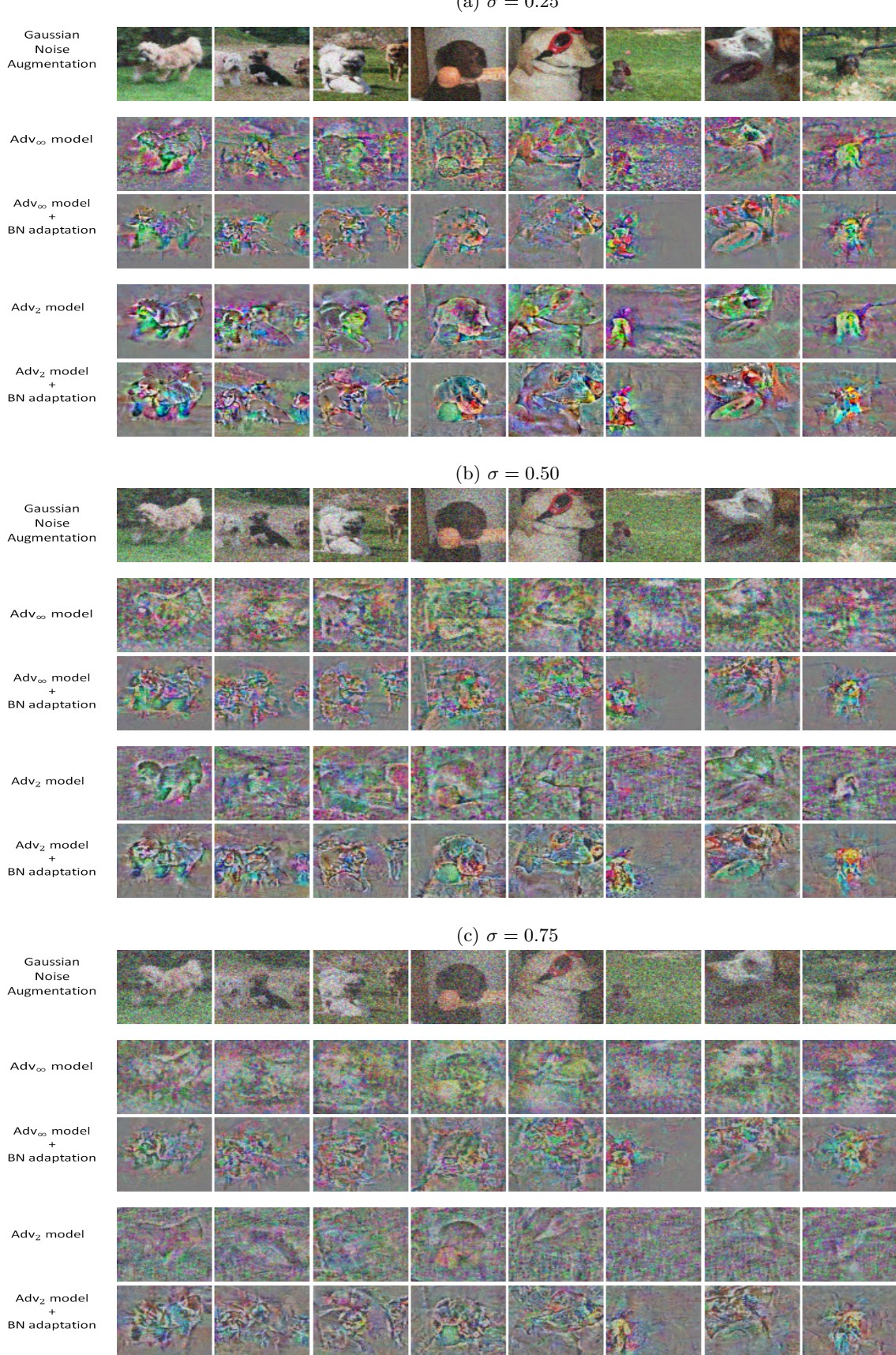

Figure 10: Additional images for visualizing loss-gradients produced by AT models at different $\sigma = \{0.25, 0.5, 0.75\}$.

