# OpenReview forum: "Towards Bridging the gap between Empirical and Certified Robustness against Adversarial Examples"
_TMLR — Rejected by TMLR_

### Review · Reviewer_KzMu · 2022-09-06

**Summary Of Contributions:**

The paper proposes to adapt a pre-trained classifier to perform well on noisy images by updating the statistics of the batch normalization layers. This allows the use of the adapted classifier to get robustness certificates via randomized smoothing. In particular, empirically robust models (obtained with adversarial training) can be integrated in the randomized smoothing framework and achieve competitive results. Moreover, the paper proposes a method, Auto-Noise, to efficiently select the best noise level for each input to obtain the largest certified radius. In the experimental evaluation on CIFAR-10 and ImageNet, Auto-Noise boosts the certificates for various classifiers.

**Broader Impact Concerns:**

None.

**Requested Changes:**

- [Critical] As mentioned above, I think the claim about having a single classifier for both empirical and certified robustness is imprecise, as the classifier is adapted (and even equipped with Auto-Noise), and its empirical robustness is not tested. I agree that the adaptation just updates a small portion of the network (but its effect on empirical robustness is unclear), but the certificates are then achieved with the smoothed classifier.

- [Critical] Please clarify the point about certification with Auto-Noise mentioned above.

- [Minor] Improving the presentation, especially adding the missing details about the method.

**Strengths And Weaknesses:**

Strengths
- Exploiting empirically robust models to improve certified robustness is an interesting approach, and the proposed adaptation method is simple and effective. This can foster new research on how to combine e.g. adversarial training (AT) with methods for certified robustness.

- The results of AT models after adaptation are competitive with those of existing methods.

- Auto-Noise further improves the certified robustness with a reasonable computational overhead.

Weaknesses
- The paper states several times that the original empirical robustness of the AT models is preserved after adaptation, obtaining both empirical and certified robustness with the same classifier. First, it seems to me that the smoothed classifiers cannot be considered the same as the original one, since its batch normalization layers are adapted, and it uses multiple forward passes with the noisy images for inference (unlike the AT model). Second, the empirical robustness of the resulting classifiers, i.e. $f_{adapt}$ (after adaptation) and $g$ (the smoothed one), is not tested.

- Auto-Noise looks like creating a set of smoothed classifiers and then selecting which one to use for each point. The certificate is computed with the selected model. In order to certify the whole pipeline (model selection plus classification), wouldn't one need to consider the robustness of the choice of the level of noise $\sigma$, since this doesn't appear to be deterministic, as well?

- I find the presentation overall a bit hard to follow. For example, there are some missing details, e.g. it's not clear which and how many images are used for adaptation (or did I miss it?). Also, some results (e.g. tables) appear far from where mentioned by the text.

---

> ### Author Response · Authors · 2022-09-07
> **Response**
>
> We thank the reviewer for their time and valuable feedback. We have addressed their comments in the updated version of our manuscript. The revised parts of the manuscript are highlighted in blue.
>
>
> - “original empirical robustness of the AT models is preserved”
>
> We would like to clarify that we intend to talk about our *Certification through adaptation using the Auto-Noise framework* for preserving the original empirical robustness. However, we agree that the empirical robustness of $f_{adapt}$ or $g$ shall be changed. We update the manuscript as follows for clarification (Sec 1):
>
> For a given test image, we first obtain the prediction using the original AT models (i.e. without adaptation). Next, we certify this predicted label using our *certification through adaptation* framework with appropriate noise-level, obtained using *Auto-Noise* technique. Hence, we maintain the same empirical robustness and benign accuracy as the existing AT models.
>
> - “the robustness of the choice of the level of noise σ”
>
> The robustness of the choice of different sigma is estimated using Eq. 5. For clarification, we rephrase the Auto-Noise technique as follows (Sec 3.3):
>
> For a given test image $x_{test}$, we estimate the certified $\ell_2$ radii for all $\sigma$ using a small set of $N_{auto}=1000$ noisy examples with high probability. While a smaller set of noisy examples produces lower certified $\ell_2$ radii, it allows us to compare the relative certified robustness achieved by different choices of $\sigma$. Hence, we select the noise level $\sigma_{auto}$ that produces the best $\ell_2$ certification using $N_{auto}$ noisy examples. Finally, we use $\sigma_{auto}$ for certifying $x_{test}$ with a large number of noisy samples, $N = 100,000$.
>
> - “which and how many images are used for adaptation ”
>
> We randomly select $1,000$ and $500$ training images to adapt the AT models for CIFAR-10 and ImageNet respectively (sec 4.2).

---

> > ### Comment · Reviewer_KzMu · 2022-09-09
> > **Follow-up questions**
> >
> > Thanks for the quick reply.
> >
> > - About the update to Sec. 1, I don't think it reflects what the proposed method does. The three classifiers $f$, $f_{adapt}$ and $g$ are distinct, and they don't need to give the same classification or being similarly robust at a test point. Then I don't think the paper can claim that "the same empirical robustness and benign accuracy as the existing AT models" is maintained without showing it. For example, for CIFAR-10, Table 1 shows that the classifier adapted from the $\text{Adv}_\infty$ model from Rice et al. (2020) achieves around 70% of clean accuracy (i.e. $l_2$ radius of 0), while the original one has above 80%.
> >
> > - A clarification about my question on the robustness of the choice of $\sigma$ in Auto-Noise. The proposed pipeline is, if I'm not missing anything, for each test point first selecting the best $\sigma$, $\sigma_{auto}$, with a small set of noisy samples and then using it with the default $N$ samples for the certification. Then the resulting certification is valid only for the chosen $\sigma_{auto}$, i.e. the probability 99.9% of the robustness of the classification is conditioned on $\sigma$. However, the algorithm for choosing $\sigma_{auto}$ is not deterministic, and a different value might be chosen if it's repeated. Then I think that the probability with which the certification holds is $0.99 \times P(\sigma = \sigma_{auto})$, and the comparison to other methods is valid only if $P(\sigma = \sigma_{auto}) = 1$, i.e. Auto-Noise is made deterministic.
> >
> > I'd like to know the authors' opinion on both points.

---

> > > ### Author Response · Authors · 2022-09-10
> > > **Response to follow-up questions:**
> > >
> > > We thank the reviewer for following-up with such engaging comments. We answer the concerns in the following and are happy to follow-up for any further comments. We shall update the paper once the reviewer approves our answers.
> > >
> > > ## Question 1
> > > We agree that “$f$, $f_{adapt}$ and $g$ are distinct, and they don't need to give the same classification or being similarly robust at a test point”. Here, we are arguing to obtain class-label prediction from $f$ as the only prediction model. *Use $f_{adapt}$ or $g$ only for certification*. More formally, for a given sample $x$ and ground-truth label $y$:
> > >
> > > - **Step 1 [Prediction]:** Return $f(x)$ as the predicted-class. Hence the benign accuracy and empirical robustness remains the same as $f$.
> > >
> > > - **Step 2 [Certification]:** Next we obtain $f_{adapt}$ and $g$ from $f$. Now, we shall certify $x$ iff  (i) $f(x)==g(x)$ and (ii) $\underline{p_A}$ > 0.5. Clearly, for a sample $x$, we may not get a certified radius.
> > >
> > > “Table 1 shows ... Adv$_{\infty}$ model ... achieves around 70% of clean accuracy ...”
> > >
> > > - In Table 1, column $\ell_2 = 0$ present the certified accuracy for $\ell_2 = 0$ using classifier $g$. However, we rely only on $f(x)$ for our predictions in the proposed framework.
> > > - Notably, we can also get $f(x)==g(x)$ by choosing $\sigma=0$. Then, we obtain the same values as the clean accuracy of $f$ at column $\ell_2 = 0$, Table 1. However, for our experiments, we choose $\sigma \in {0.12, 0.25, 0.37, 0.50, 0.67, 0.75, 0.87, 1.0}$.
> > >
> > > - Please note: Table 1 reports the results using the best training hyper-parameters. Please refer to Table 4 (Appendix) for results of Adv$_{\infty}[8/255]$ +Adaptation +Auto-Noise that produces accuracy=79.69% at $\ell_2=0$.
> > >
> > > ## Question 2:
> > > “ 0.999×P(\sigma= \sigma_{auto})” is the *joint probability* of producing the certified $\ell_2$ radius using our Auto-Noise method. However, we are interested in certifying the $\ell_2$ radius, and not the joint-probability. Let us explain it as follows:
> > >
> > > **Notations:** For a given sample, $x$, let $R$ denote the maximum certified radius. The randomized smoothing technique can only estimate the lower-bounds of R (hence the process is not deterministic). Let $\{ \sigma_1, \cdots, \sigma_k \}$ is the set of noises for our Auto-Noise. $R_{\sigma_i, N_{auto}}$ and $R_{\sigma_i, N}$ denotes the certified radius obtained using noise-level $\sigma_i$ and total noisy samples, $N_{auto}$ and $N$ respectively.
> > >
> > >
> > > Then the *joint probability* of obtaining $R_{\sigma_i, N}$ using Auto-Noise technique is:
> > >
> > > $\mathcal{P}([R_{\sigma_i, N_{auto}} \geq R_{\sigma_j, N_{auto}} \forall \sigma_j ] and [R \geq R_{\sigma_i, N}])$
> > >
> > > = $\mathcal{P}([R_{\sigma_i, N_{auto}} \geq R_{\sigma_j, N_{auto}} \forall \sigma_j ]) \times \mathcal{P}([R \geq R_{\sigma_i, N}] | [R_{\sigma_i, N_{auto}} \geq R_{\sigma_j, N_{auto}} \forall \sigma_j ])$
> > >
> > > =  $\mathcal{P}(\sigma_i) \times \mathcal{P}([R \geq R_{\sigma_i, N}] | \sigma_i) $
> > >
> > > = $\mathcal{P}(\sigma_i) \times 0.999 $
> > >
> > > However, we are interested to ensure that the certified radius is guaranteed with 99.9% probability i.e $\mathcal{P}([R \geq R_{\sigma_i, N}] = 0.999$; which is ensured by the Randomized Smoothing algorithm. For a better clarification, let us consider the *brute-force* version of Auto-Noise as follows:
> > >
> > > **[Brute-Force version of Auto-Noise]**
> > > we compute ${R_{\sigma_1, N} \cdots R_{\sigma_K, N}}$ *independently* for all $\sigma_j$ with 99.9% probability and return the maximum certified radius, denoting as $R_{\sigma_k, N}$. This process *removes the selection bias* for $\sigma$ in this process. Now,
> > >
> > > $\mathcal{P}([R \geq \max {R_{\sigma_1, N} \cdots R_{\sigma_K, N}}]) = \mathcal{P}([R \geq R_{\sigma_k, N}]) = 0.999$
> > >
> > > Therefore, since $R_{\sigma_k, N} \geq R_{\sigma_i, N}$, the certified radius $R_{\sigma_i, N}$ always satisfies $\mathcal{P}([R \geq R_{\sigma_i, N}] = 0.999$.
> > >
> > > In other words, $\mathcal{P}([R \geq R_{\sigma_i, N}] | \sigma_i)  = \mathcal{P}([R \geq R_{\sigma_i, N}]) = 0.999$ i.e., independent of $\sigma_i$. However, whether the Auto-Noise would produce $R_{\sigma_i, N}$ as the certified radius is dependent on $\sigma_i$.

---

### Review · Reviewer_JhFy · 2022-09-08

**Summary Of Contributions:**

This paper proposes an adapted randomized smoothing certification methods based on two simple yet effective approaches: (1) adapting BN according to train/test samples against Gaussian perturbations and (2) using auto-noise to select appropriate noise levels for best certification performance. The empirical results show the practical effectiveness of the both approaches and the combination of both techniques on L2-adversarially trained models achieves the best ACR (average certified radius).

**Broader Impact Concerns:**

Broader impact statement is included in paper. No concern remains.

**Requested Changes:**

### Changes
1. The background and introduction described both randomized smoothing style and deterministic certification methods. The paper emphasizes the scalability of the first category, which is reasonable. However, the benefits of deterministic certification methods as well as the limitations of randomized smoothing should also be mentioned. Otherwise, it leaves a wrong impression that randomized smoothing style certification is always better, which is not true. Recently, people has made great progress in scaling up deterministic certification methods like the SOTA winning tool alpha-beta-CROWN [A,B] and other participants like VeriNet [C, D] and OVAL [E, F] in VNN-COMP22. They should also be appreciated in the related works.
2. It seems a little odd to put randomized smoothing background in section 3.
3. SOTA certifiable defenses like CROWN-IBP [G] and especially COLT [H] that aims to close the gap between adversarial and provable robustness could also be mentioned in related works.

[A] Fast and Complete: Enabling Complete Neural Network Verification with Rapid and Massively Parallel Incomplete Verifiers", ICLR’21

[B] Beta-CROWN: Efficient Bound Propagation with Per-neuron Split Constraints for Complete and Incomplete Neural Network Verification, NeurIPS’21

[C] Efficient Neural Network Verification via Adaptive Refinement and Adversarial Search, ECAI’20

[D] DEEPSPLIT: An Efficient Splitting Method for Neural Network Verification via Indirect Effect Analysis, IJCAI’21

[E] Scaling the Convex Barrier with Sparse Dual Algorithms, ICLR’21

[F] Improved Branch and Bound for Neural Network Verification via Lagrangian Decomposition, arXiv

[G] Towards Stable and Efficient Training of Verifiably Robust Neural Networks, ICLR’20

[H] Adversarial Training and Provable Defenses: Bridging the Gap, ICML’20

**Strengths And Weaknesses:**

### Strength
- The two proposed approaches are simple yet effective in practice.
- It is non-trivial to achieve empirically better ACR than SOTA randomized smoothing and SmoothAdv.

### Weakness
- The novelty is quite limited for each proposed approach but seemingly not to be a big concern considering conciseness and appropriate empirical analysis.
- The proposed approaches especially adaptation BN are not that effective on original randomized smoothing/SmoothAdv models. How to balance between proposed adaptation approach, adversarial robustness, and robustness against noise in the training process to best fit the test-time certification seems to be a very interesting direction to investigate.

---

> ### Author Response · Authors · 2022-09-10
> **Response**
>
> We thank the reviewer for their time and valuable feedback. We have addressed their comments in the updated version of our manuscript. The revised parts of the manuscript are highlighted in orange.
>
> - [1.] We have included the suggested references in Section 2.1.2
> - [2.] We moved the entire randomized smoothing background from Section 3 to 2.1.3
> - [3.] References for  CROWN-IBP and COLT is added in section 2.1.2.

---

### Review · Reviewer_Mjv7 · 2022-09-17

**Summary Of Contributions:**

This paper proposes using adaptive batch norm to improve the certification radius of the randomized smoothing technique. To choose a suitable noise level $\sigma$, it proposes Auto-Noise technique wherein noise levels in a grid are compared based on certification levels with $N_{auto} = 1000$.

**Broader Impact Concerns:**

I have no concern on  the ethical implications of the work.

**Requested Changes:**

1) Please rewrite Section 3 and Algorithm 1 to describe the proposed approach better and more clearly.
2) The setting of adaptive batch norm is unclear. The authors mention that $\rho$ is used to balance between the training set and test batch statistics. It is unclear if in the experiments, a test batch or a single test example is certified at a time and how to set $\rho$.
3) The paper highlights texts using orange color which makes it harder to read.
4) How can you rely on the estimation of a certification level (e.g., a certification radius) using only $1,000$ examples to compare various noise levels? In the paper of randomized certification, it requires $100,000$ examples to estimate this quantity.
5) How to compute Top 1 accuracy in Table 1? Is it top 1 accuracy on benign images, Gaussian noised images, or adversarial examples?

**Strengths And Weaknesses:**

Strengths
- Literature review section is rich.

Weaknesses
- The novelty of this paper is very limited. The main contribution is to leverage the adaptive batch norm technique with the randomized smoothing technique. The adaptive batch norm technique has been widely applied in domain adaptation, while the randomized smoothing technique has been proposed before.
- The Auto-noise technique is heuristic. There is nothing to guarantee the certification level estimated using only $1,000$ examples is trustable to compare various noise levels. Additionally, Auto-noise is naturally a grid search via certification level estimated with $1,000$ examples.
- The writing is not good especially Section 3. In Algorithm 1, there are several strange notations such as $CLONE(f.train())$, $f_{adapt}.eval()$.
- The experimental results are humble. It is hard to convince the advantages of the proposed approach. For example, in Figure 3, the baseline $RAND_{\sigma=0.5}$ outperforms the proposed approach.

---

> ### Author Response · Authors · 2022-09-19
> **Response**
>
> We thank the reviewer for their time and valuable feedback. We have addressed their comments in the updated version of our manuscript. The revised parts of the manuscript are now highlighted in Blue for better readablity.
>
> [W]: Weaknesses     [RC] Requested Changes
>
>
> [W1] “novelty of this paper is very limited.”
>
> - **Ans.** To the best of our knowledge, we are the first to produce certified robustness at large $\ell_2$  radii for AT model for large-scale networks (e.g. ResNet) and datasets (e.g. ImageNet). We claim it as the novel and significant contribution as in sensitive real-world applications, we need both certified and empirical robustness. (Restated in Section 1, Contribution 1)
>
> [W3, RC 1]. We have rewrote Section 3 and Algorithm 1.
>
> - **Ans.** We have rewritten Section 3 and Algorithm 1 and removed the mentioned notations.
>
> [RC 2] ”It is unclear if in the experiments, a test batch or a single test example is certified at a time and how to set ρ”
>
> - **Ans.** We use a set of clean training images, $X_{clean}$ and apply full-adaptation using ρ=1. (Updated in Section 3).
>
> [RC 3] The paper highlights texts using orange color which makes it harder to read.
>
> - **Ans.** We now highlight the revised parts of the manuscript in “Blue” for better readability.
>
> [W2, RC4] “How can you rely on the estimation of a certification level (e.g., a certification radius) using only 1,000 examples to compare various noise levels? In the paper of randomized certification, it requires 100,000 examples to estimate this quantity.”
>
> - **Ans.** Please note that randomized smoothing does not require 100,000 samples to produce $\ell_2$ certified radii. *Even a smaller set of noisy examples (e.g. $N_{auto}=1000$) can provide certification with high probability (e.g. 99.9\% confidence), however, for smaller $\ell_2$ certified radii.* Hence, we can still fairly compare among different noise-levels. (Updated in Section 3.2).
>
> - The original randomized smoothing paper also presented the $\ell_2$ certification using 1000 noisy examples and $99.9\%$ confidence. (See Figure 8 [middle] of Cohen et al. (2019) [https://arxiv.org/pdf/1902.02918.pdf]).
>
> [W4] “It is hard to convince the advantages of the proposed approach. Figure 3, the baseline RANDσ=0.5 outperforms the proposed approach.”
>
> - **Ans.** Figure 3 only demonstrates non-trivial robustness for AT models using fixed-noise and smaller threat-boundaries.
>
> - Figure 4 shows that as we train AT models with larger threat-boundaries, we achieve significantly better performance and out-performs RANDσ=0.5. We further improve this performance using Auto-Noise in Figure 5. Please refer to Table 3 and 4 for detailed results on ImageNet and CIFAR-10 respectively.
>
> - In summary, we produce ACR scores up to $1.102$ and $1.148$ for CIFAR-10 and ImageNet using AT models. We also provide the state-of-the-art performance for CIFAR-10. (detailed results are provided in Table 3 and 4 in Appendix).
>
>
> [RC 5] How to compute Top 1 accuracy in Table 1? Is it top 1 accuracy on benign images, Gaussian noised images, or adversarial examples?
>
> - **Ans.** In Table 1, the $\ell_2$ radius 0 represents the accuracy for Gaussian noised images i.e. obtained from the smoothed classifier $g$. This is different from the benign accuracy of our proposed framework. (We have included an additional Table 2 for clarification).

---

### Review · Reviewer_cfGb · 2022-09-18

**Summary Of Contributions:**

This paper proposes a certification method for adversarially robust models with empirical robustness without a certificate. Their proposed method has two parts, the first part is a model adaptation method that given a Gaussian noise level, adapts a model for certified robustness at that level. The second part, Auto-Noise, finds an optimal Gaussian noise level per test sample.

More specifically, the BN-adaptation (Section 3.2), takes a pretrained model, particularly an adversarially trained model and trains only the batch normalization layers on Gaussian-perturbed inputs with a pre-selected noise level. Figure 1 argues, adapting the model on the train dataset provides similar performance to adapting on the test set. So the method doesn’t require unseen data for adaptation.

**Broader Impact Concerns:**

No concern.

**Requested Changes:**

Please see the "Strengths and Weaknesses" section.

**Strengths And Weaknesses:**

Strengths:
- Results in Table 1 show a boost in the ACR for the proposed method.
- The proposed method has relatively low cost of adaptation and can be applied generically to any robust model with empirical robustness.
- Fig 2 supports the claim that BN-adapted models have more visually discriminative gradients wrt the input, particularly at high noise ratio (sigma=0.75). Can authors provide additional examples in the appendix? This figure shows only two inputs.

Questions:
- “Effect on empirical robustness and benign (clean) accuracy”: this paragraph does not support the claim by referencing a table or figure. Please provide empirical support for this claim.
- Auto-Noise: can you clarify if you use sigma_auto only for certification in Line 6 of Algorithm 1 or also for BN-adaptation in Line 1 of Algorithm 1? From the method names in Table 1, it seems the second one is intended. I suggest also using two different notations instead of one sigma in Algorithm 1 for both BN-adaptation and certification. It would also help if Algorithm 1 is updated with the steps of Auto-Noise, possibly in a different color. If this is the correct interpretation, would it improve the performance if after BN-adaptation and Auto-Noise, the sigma_auto is used in another BN-adaptation step?
- Table 2: Wouldn't it be possible to compare with SmoothAdv on ImageNet as well? Table 1 includes the comparison on CIFAR-10. Can you clarify in the caption whether the baseline is Rand0.5?
- Can you clarify how the certified robustness plots are generated for a randomized smoothing-based method with a pre-selected sigma? Do the plots show the frequency of the number of test samples with certified sigma larger than a given value on the x-axis? Or do you compute the accuracy by varying the sigma used in the function g? I doubt it’s the second but just wanted a clarification.
- Fig 3: is the emphasis on the “non-trivial” accuracy? Because the proposed method is not necessarily the best method compared to prior work.
- Fig 4a,b: The proposed method seem to show different tradeoffs at different values of sigma. Is the best method chosen best on higher ACR value?
- Fig 5: Can you report the sigma found using Auto-Noise and the difference in the value from the sigma used for BN-adaptation for different models?
- Fig 6: Can the authors add the BN-adapt method without Auto-Noise to these plots?
- Fig 6b,c: Auto+Adapt methods in these plots achieve lower ACR than Auto method. Can you explain why?
- Table 3: what is the Baseline if it is not Rand0.5? The experiment section says the baseline is always Rand0.5 but Table 3 shows rows for both the Baseline and Rand0.5 and the values are different.

---

> ### Author Response · Authors · 2022-09-19
> **Response**
>
> We thank the reviewer for their time and valuable feedback. We have addressed their comments in the updated version of our manuscript. The revised parts of the manuscript are highlighted in blue.
> [S]: Strengths   [Q]: Questions
>
> [S1] “Can authors provide additional examples in the appendix? This figure shows only two inputs.”
>
> - **Ans**. Additional results with 10 images are included in Figure 8 (page 22).
>
> [Q1] “Effect on empirical robustness and benign (clean) accuracy… empirical support for this claim.”
>
> - **Ans.** We have included an additional Table 2 for this explanation.
>
> [Q2] “Auto-Noise … clarification”
>
> - **Ans.** Yes. We are using “the sigma_auto is used in another BN-adaptation step” to obtain $f_{adapt}$, followed by executing the randomized-smoothing algorithm using $\sigma_{auto}$. We include Algorithm 2 for clarification.
>
> [Q3] “Can you clarify in the caption whether the baseline is Rand0.5?”
> [Q10] Table 3: what is the Baseline if it is not Rand0.5?
>
> - **Ans.**  Baseline is the *standard, non-robust DNN classifiers, trained using clean images*. (Updated captions in Figure 3, Table 4,5 [Appendix]). Rand_0.5 are trained using Gaussian noise $(\sigma=0.5)$.
>
> Also, we have now simplified Table 1 by removing the comparative results for better readability. (Table 4,5 [Appendix] presents the detailed comparative results).
>
> [Q4] “how the certified robustness plots are generated”
>
> - **Ans.** Yes. “the plots show the frequency of the number of test samples with certified sigma larger than a given value on the x-axis”.
>
> [Q5] “Fig 3: is the emphasis on the “non-trivial” ”
>
> - **Ans.** We have included the explanation for using the term “non-trivial”:
> *We mainly compare with the non-adapted AT models that produce ``trivial'' performance i.e., very small $\ell_2$ certified accuracy. Compared to that, we achieve significantly better results using “certification through adaptation” method.*
>
> [Q6] “Fig 4a,b… Is the best method chosen based on higher ACR value?”
>
> - **Ans.** Yes
>
> [Q7] “Fig 5: Can you report the sigma found using Auto-Noise and the difference in the value from the sigma used for BN-adaptation for different models?”
>
> - **Ans.** We have added Figure 8, 9 in Appendix A.2 to demonstrate the sigma_auto produced by Auto-Noise for different models.
>
> [Q8] “Fig 6: Can the authors add the BN-adapt method without Auto-Noise to these plots?”
>
> - **Ans.** We have updated Figure 6 including the BN-adapt method without Auto-Noise, denoted as (Adapt at $\sigma=0.5$).
>
> [Q9] “6b,c: Auto+Adapt methods in these plots achieve lower ACR than Auto method.”
>
> - **Ans.** We have included the following explanation in “Certification through Adaptation for existing models” (See Page 11): *The existing randomized-smoothing models are already trained using Gaussian noise. Hence, further improvement in robustness against different levels of Gaussian noise for a given image becomes challenging by only incorporating BN adaptation techniques.*

---

### Review · Reviewer_xKY5 · 2022-09-19

**Summary Of Contributions:**

This paper provides a robustness certification method for off-the-shelf networks. It uses randomized smoothing to provide such certificates. Since randomized smoothing demands invariance to Gaussian noise, it adapts batch-norm statistics in the off-the-shelf network to instill such invariance, demonstrating that covariates shift induced by Gaussian noise in the input is easily correct by simply updating normalization statistics. Overall this work unifies certified and empirical robustness, with a straightforward approach to obtaining certificates of empirical robust predictions.

**Requested Changes:**

Changes are already highlighted in aforementioned comments.

**Strengths And Weaknesses:**

This approach is an off-the-shelf method to certify the prediction of non-robust/robust neural networks. Surprisingly existing empirically robust classifiers can be certified with a very high score, i.e., very high certified robust accuracy along with strong empirical robust accuracy. However, the applicability of the method to off-the-shelf nets would be better demonstrated by directly testing on such networks
- I encourage authors to test their methods on state-of-the-art off-the-shelf models available from RobustBench [1] (it provides more than 50 different networks on cifar10)
- For imageNet authors can use networks from RobustBench or other latest repositories [2, 3].

These sota networks with very high empirical robust accuracy may help this work works to also achieve sota-certified robust accuracy.


*Limitation to batch-norm*: This is a key limitation of existing work and is hard to ignore. Adapting batch-norm is reasonable, as it has been widely used. However, in the modern class of networks, BN is quickly being replaced with LayerNorm or other instance-based normalization techniques. E.g., the vision transformer predominantly uses LayerNorm instead of batch-norm. Even modern CNNs, e.g., ConvNext [6], have dropped batchnorm in favor of layernorm. I believe a more futuristic method would not limit to BN but also consider going adapting to other layers. E.g., even other layers, such as convolutions, can be updated slightly with a low learning rate to account for the drift of activations under noise in the input.


*Average robustness*: Fig. 3a show an interesting trend. While the average robustness of the proposed approach is competitive its certified robust accuracy compared to Rand_0.5 models is poor. It appears the design choices for the proposed methods are catered for high average case robustness since at high perturbation budgets proposed method achieves better results. If we are concerned about robustness on a specific radius (e.g., the radius of 0.5 of ImageNet), rather than average, then would the method do better?

On a presentation front, I recommend using a distilled version of Table 1 in the introduction and keeping the full table in the experimental result section. To a reader who is not familiar with this research domain, it would either requires an extensive explanation of the setup (e.g., what are the baselines, distinguishing between proposed and previous work)

1. https://robustbench.github.io/
2. https://github.com/dedeswim/vits-robustness-torch
3. https://github.com/alibaba/easyrobust
4. https://github.com/rwightman/pytorch-image-models/blob/master/timm/models/vision_transformer.py
5. https://github.com/facebookresearch/xcit/blob/main/xcit.py
6. Liu, Zhuang, et al. "A convnet for the 2020s." Proceedings of the IEEE/CVF Conference on Computer Vision and Pattern Recognition. 2022. https://github.com/facebookresearch/ConvNeXt/blob/main/models/convnext.py

---

> ### Author Response · Authors · 2022-09-19
> **Response**
>
> We thank the reviewer for their time and valuable feedback and liking this work. We have addressed their comments in the updated version of our manuscript. The revised parts of the manuscript are highlighted in blue.
>
> [Q1] “the applicability of the method to off-the-shelf nets would be better demonstrated by directly testing on such networks”
>
> - **Ans.**
> Yes. We also believe that we can further improve the certification performance by using more recent AT methods with higher empirical robustness.
> However, note that our proposed method requires the AT models to be trained at different threat boundaries to fairly evaluate and understand the certification performances (see Figure 4). Hence, we need to retrain multiple copies of these different models along with computing certification results to provide such comparisons. Since it requires a significant amount of computational resources, we could not include these experiments .
> We agree that understanding the interplay between their empirical robustness and certification against adversarial examples can be an interesting future work for both research and practical applications. (Updated in our conclusion)
>
>
> [Q2] “Limitation to batch-norm: ”
>
> - **Ans.** For modern networks that have replaced BN layers, we can investigate other adaptation techniques, e.g., self-supervised domain adaptation on single images, pseudo-labeling, entropy-minimization etc. (We have included this in our updated conclusion)
>
>
> [Q3] “Average robustness:”
>
> - **Ans.** We can empirically check the appropriate threat-boundaries to find the best AT model can be used for each use-case. For example, in Table 5 (Appendix), we observe that AT models, trained using lower threat boundaries, produce higher robustness for smaller $\ell_2$ radii (and vice-versa). For example, Adv$_2[1]$ produces the best certification at $\ell_2=0.5$ for CIFAR-10.
>
> [Q4] “distilled version of Table 1”:
>
> - **Ans.** We have updated Table 1, keeping only the results for AT models according to your suggestion.

---

### Author Response · Authors · 2022-09-16
**Repost to the follow-up questions by Reviewer KzMu**

Dear Reviewer KzMu, we just realized that our reply to your follow-up question was not visible to you (and we are not able to edit the visibility). Hence, we are re-posting the answers below.

We thank you for following-up with such engaging comments. We answer the concerns in the following and are happy to follow-up for any further comments. We shall update the paper once the reviewer approves our answers.

## Question 1
We agree that “$f$, $f_{adapt}$ and $g$ are distinct, and they don't need to give the same classification or being similarly robust at a test point”. Here, we are arguing to obtain class-label prediction from $f$ as the only prediction model. *Use $f_{adapt}$ or $g$ only for certification*. More formally, for a given sample $x$ and ground-truth label $y$:

- **Step 1 [Prediction]:** Return $f(x)$ as the predicted-class. Hence the benign accuracy and empirical robustness remains the same as $f$.

- **Step 2 [Certification]:** Next we obtain $f_{adapt}$ and $g$ from $f$. Now, we shall certify $x$ iff  (i) $f(x)==g(x)$ and (ii) $\underline{p_A}$ > 0.5. Clearly, for a sample $x$, we may not get a certified radius.

“Table 1 shows ... Adv$_{\infty}$ model ... achieves around 70% of clean accuracy ...”

- In Table 1, column $\ell_2 = 0$ present the certified accuracy for $\ell_2 = 0$ using classifier $g$. However, we rely only on $f(x)$ for our predictions in the proposed framework.
- Notably, we can also get $f(x)==g(x)$ by choosing $\sigma=0$. Then, we obtain the same values as the clean accuracy of $f$ at column $\ell_2 = 0$, Table 1. However, for our experiments, we choose $\sigma \in {0.12, 0.25, 0.37, 0.50, 0.67, 0.75, 0.87, 1.0}$.

- Please note: Table 1 reports the results using the best training hyper-parameters. Please refer to Table 4 (Appendix) for results of Adv$_{\infty}[8/255]$ +Adaptation +Auto-Noise that produces accuracy=79.69% at $\ell_2=0$.

## Question 2:
“ 0.999×P(\sigma= \sigma_{auto})” is the *joint probability* of producing the certified $\ell_2$ radius using our Auto-Noise method. However, we are interested in certifying the $\ell_2$ radius, and not the joint-probability. Let us explain it as follows:

**Notations:** For a given sample, $x$, let $R$ denote the maximum certified radius. The randomized smoothing technique can only estimate the lower-bounds of R (hence the process is not deterministic). Let $\{ \sigma_1, \cdots, \sigma_k \}$ is the set of noises for our Auto-Noise. $R_{\sigma_i, N_{auto}}$ and $R_{\sigma_i, N}$ denotes the certified radius obtained using noise-level $\sigma_i$ and total noisy samples, $N_{auto}$ and $N$ respectively.


Then the *joint probability* of obtaining $R_{\sigma_i, N}$ using Auto-Noise technique is:

$\mathcal{P}([R_{\sigma_i, N_{auto}} \geq R_{\sigma_j, N_{auto}} \forall \sigma_j ] and [R \geq R_{\sigma_i, N}])$

= $\mathcal{P}([R_{\sigma_i, N_{auto}} \geq R_{\sigma_j, N_{auto}} \forall \sigma_j ]) \times \mathcal{P}([R \geq R_{\sigma_i, N}] | [R_{\sigma_i, N_{auto}} \geq R_{\sigma_j, N_{auto}} \forall \sigma_j ])$

=  $\mathcal{P}(\sigma_i) \times \mathcal{P}([R \geq R_{\sigma_i, N}] | \sigma_i) $

= $\mathcal{P}(\sigma_i) \times 0.999 $

However, we are interested to ensure that the certified radius is guaranteed with 99.9% probability i.e $\mathcal{P}([R \geq R_{\sigma_i, N}] = 0.999$; which is ensured by the Randomized Smoothing algorithm. For a better clarification, let us consider the *brute-force* version of Auto-Noise as follows:

**[Brute-Force version of Auto-Noise]**
we compute ${R_{\sigma_1, N} \cdots R_{\sigma_K, N}}$ *independently* for all $\sigma_j$ with 99.9% probability and return the maximum certified radius, denoting as $R_{\sigma_k, N}$. This process *removes the selection bias* for $\sigma$ in this process. Now,

$\mathcal{P}([R \geq \max {R_{\sigma_1, N} \cdots R_{\sigma_K, N}}]) = \mathcal{P}([R \geq R_{\sigma_k, N}]) = 0.999$

Therefore, since $R_{\sigma_k, N} \geq R_{\sigma_i, N}$, the certified radius $R_{\sigma_i, N}$ always satisfies $\mathcal{P}([R \geq R_{\sigma_i, N}] = 0.999$.

In other words, $\mathcal{P}([R \geq R_{\sigma_i, N}] | \sigma_i)  = \mathcal{P}([R \geq R_{\sigma_i, N}]) = 0.999$ i.e., independent of $\sigma_i$. However, whether the Auto-Noise would produce $R_{\sigma_i, N}$ as the certified radius is dependent on $\sigma_i$.

---

### Decision · Action_Editors · 2022-11-06

**Recommendation:** Reject

**Comment:**

The approach is interesting, but as the reviewers pointed out, it has certain limitations. While seemingly based on sound principles, the proposed methodology is mostly heuristic. There is no theory to support the claims here, even in simpler settings. This makes it hard to have a deeper understanding and appreciation of the proposed methodology. The approach is based on an existing mechanism called adaptive batch normalization. The other component of the approach, i.e., randomized smoothing, is also prior art and well-understood. Reviewers also point out that the method is limited to networks with batch normalization and that the empirical results are somewhat lacking and unconvincing.

Based on my reading of the paper, comments from the reviewers, and careful consideration of the authors' responses, my assessment is that the paper is not yet ready for publication in TMLR; the changes suggested by the reviewers would require a significant revision. I encourage the authors to incorporate the feedback from the reviewers in working toward a stronger submission.

**Audience:**

Researchers working on adversarial robustness, in particular, with emphasis on providing robustness certificates/guarantees.

**Claims And Evidence:**

The paper proposes a method for transforming an adversarially trained model into a randomized smoothed classifier. The authors claim that the proposed method achieves the same empirical robustness as the original adversarially trained model but enjoys certified robustness. Empirical results on CIFAR and ImageNet are presented to support the claims.